# DataGuard: A Non-intrusive Dataset Auditing Framework via Differential Information Forensics

**Jiadong Lou** [1]   **Wenxin Rong** [2]   **Li Chen** [3]   **Xing Gao** [2]   **Rui Zhang** [2]   **Xu Yuan** [2]

## Abstract

Concerns over dataset misuse in deep learning have highlighted the need for effective auditing. Unlike existing intrusive methods that require dataset modifications, which risk model performance and security, we present DataGuard, a non-intrusive framework for quantitative dataset auditing. Specifically, DataGuard integrates three key components: 1) a differential comparison between the target dataset and auxiliary non-training datasets, 2) an information-forensic analysis establishing formal inequalities to distinguish training data; and 3) a multivariate statistical test that translates these discrepancies into rigorous auditing scores. Extensive experiments demonstrate that DataGuard can detect both full and partial dataset usage while remaining robust under diverse practical training scenarios, offering a principled, information-theoretic solution for transparent AI development.

## 1. Introduction

The remarkable development of advanced deep learning models has spurred their rapid deployment for commercial use, supporting services ranging from image and video generation to text-based interactions. Given the data-driven nature of these systems, training requires massive, diverse, and unique datasets, driving providers to engage in extensive data collection. However, in the absence of effective auditing mechanisms, these practices have raised growing concerns regarding dataset misuse. Indeed, the unauthorized use of proprietary data has triggered high-profile lawsuits, such as visual artists suing Midjourney (Sue, c;b) and The New York Times taking legal action against OpenAI (Sue, a). Additionally, concerns have arisen regarding the com-

mercial exploitation of academic datasets. Despite these pressing needs, a robust methodology for dataset auditing remains elusive.

Predominant auditing approaches (Li et al., 2023b; Guo et al., 2023; Tang et al., 2023; Li et al., 2022; Sablayrolles et al., 2020; Wang et al., 2024; Wenger et al., 2024; Chen & Pattabiraman, 2024; Dong et al., 2023; Du et al., 2024; Huang et al., 2024; Liu et al., 2022; Huang et al., 2022) are dataset-intrusive, where data owners deliberately inject watermarks or perturbations into datasets prior to release. Then, models trained on the watermarked data will exhibit distinctive prediction patterns, thereby enabling dataset auditing in a manner analogous to verifying backdoor attacks or membership inference attacks. While effective for verification, these intrusive modifications inevitably compromise model performance and introduce substantial security risks. As highlighted in (Guo et al., 2023), even carefully designed "harmless" watermarks can lead to erroneous predictions on a considerable portion (10%) of the data. Furthermore, such modifications must be injected before dataset publication, rendering these methods ineffective for protecting the vast number of datasets that have already been released.

Conversely, recent studies (Du et al., 2025) have explored adapting dataset inference methods (Xu & Li, 2022; Maini et al., 2021; Dziedzic et al., 2022; Dong et al., 2023), which were originally designed for copyright protection, for non-intrusive auditing. However, these approaches rely on discrepancies of empirical signals (e.g., accuracy, confidence, decision boundary gaps) between training and non-training datasets. As noted in (Du et al., 2025), these empirical metrics lack a principled mathematical connection to the training process and are highly sensitive to training randomness (e.g., noise, regularization), rendering them unreliable for rigorous auditing. These practical limitations underscore the necessity of developing a mathematically grounded non-intrusive auditing method.

In this paper, we propose a novel non-intrusive framework, named DataGuard, for dataset auditing. The primary goal of DataGuard is to provide a quantitative auditing score that reflects the likelihood of a suspect classification model using the target dataset for the training process. Moreover, DataGuard is designed to preserve the integrity of datasets,

---

[1]Rowan University [2]University of Delaware [3]University of Louisiana at Lafayette. Correspondence to: Xu Yuan <xyuan@udel.edu>.

*Proceedings of the 43rd International Conference on Machine Learning*, Seoul, South Korea. PMLR 306, 2026. Copyright 2026 by the author(s).

require no modifications to the target dataset, and leverage interpretable indicators moving beyond empirical observations under the black-box setting. These commitments introduce three key challenges: 1) Integrity: extracting evidence intrinsic to the training process without modifying the dataset, 2) Interpretability: employing mathematically grounded rather than heuristic signals, and 3) Auditability: producing reliable scores that avoid false positives while remaining rigorous.

To address these challenges, we rely on three key insights: differential comparison, information forensics, and multivariate statistical test. First, the success of dataset auditing hinges on the ability to distinguish training data from non-training data, which motivates us to compare the target dataset to non-training counterparts. Given that published datasets are often curated from large raw data collections, the portions untouched and unexposed to model providers can be constructed as auxiliary datasets to provide non-training dataset baselines. Second, once a model has been trained on a dataset, it inevitably retains information about the data samples within that dataset. To measure this retention, we employ information forensics based on the Information Plane (Shwartz-Ziv & Tishby, 2017) to trace how much information the model predictions preserve about the input data and ground-truth labels. Crucially, rather than treating information as empirical indicators, we establish formal inequalities on the information plane that mathematically separate training datasets from non-training datasets, thereby providing a principled foundation for distinguishing whether the suspect model has used the target dataset. Finally, to translate the discrepancies of information metrics between the target and auxiliary datasets into a quantitative auditing score, DataGuard leverages a multivariate statistical test, i.e., Hotelling $T^2$ testing (Hotelling et al., 1931). Based on this testing, DataGuard produces a statistical $p$-value as the auditing score, providing interpretable and quantitative evidence of dataset usage.

Extensive experiments are conducted to evaluate the performance of our DataGuard across different datasets and model structures. Our results demonstrate that DataGuard identifies all models trained on target datasets under different training settings and outperforms existing dataset-intrusive techniques. Moreover, DataGuard also demonstrates accurate auditing performance in cases where only a partial target dataset is used to train the suspect model or the target dataset constitutes only a portion of the suspect model's training data.

Our contributions can be summarized as follows: 1) we present a novel non-intrusive dataset auditing framework that avoids any dataset modifications; 2) we establish formal information-theoretic inequalities that separate training datasets from non-training counterparts, providing a princi-

pled foundation for future auditing research; 3) we provide statistically defensible evidence of dataset usage, advancing transparency in AI system development.

## 2. Related Work and Motivation

### 2.1. Existing Auditing Methods

**Watermarking-based Auditing.** Early approaches to dataset auditing primarily relied on watermarking-based techniques (Li et al., 2023b; Guo et al., 2023; Tang et al., 2023; Li et al., 2022; Sablayrolles et al., 2020; Wang et al., 2024; Wenger et al., 2024; Chen & Pattabiraman, 2024; Dong et al., 2023; Du et al., 2024). These methods embed owner-specific patterns into a dataset before its release, such that any model trained on the watermarked dataset will exhibit corresponding prediction behaviors, serving as evidence of dataset usage. However, watermarking is inherently intrusive as it requires modifying the dataset and inducing artificial behaviors in the trained model. Even clean-label, untargeted, and domain-specific variants (Li et al., 2023b; 2022; Guo et al., 2023) still rely on backdoor mechanisms and may introduce security vulnerabilities.

**Membership Inference–based Auditing.** Another line of work uses membership inference (MI) as a signal for dataset auditing (Huang et al., 2024; Liu et al., 2022; Huang et al., 2022). While membership inference is naturally suited for per-sample auditing, extending it to the dataset level requires introducing additional modifications to the dataset. For instance, (Huang et al., 2024) perturbs samples in the original dataset to produce two versions, releasing only one to the public. By evaluating whether a suspect model shows different inference accuracy across these versions, data owners can detect the unauthorized training.

**Dataset Inference for Copyright Protection.** Dataset inference methods (Xu & Li, 2022; Maini et al., 2021; Dziedzic et al., 2022; Dong et al., 2023), originally developed for model copyright protection, have recently been discussed as potential non-intrusive auditing tools (Du et al., 2025). These methods evaluate whether a probing dataset is similar to the training dataset of a suspect model by comparing empirical behavioral signals such as prediction accuracy, confidence, or decision-boundary distance. However, as noted in (Du et al., 2025), these metrics are sensitive to randomness in the training process, thus remain unstable for real-world auditing.

In summary, existing techniques either rely on dataset modifications or empirical signals, underscoring the need for a theoretically grounded non-intrusive auditing method.

### 2.2. Challenges and Key Intuitions

Given the limitations of existing methods, developing a non-intrusive auditing framework requires overcoming three

fundamental hurdles.

**Integrity.** The primary challenge in non-intrusive auditing is distinguishing training patterns without modifying the dataset. A plausible intuition is to compare the suspect model against a locally trained "shadow model," assuming that training on the same data yields aligned behaviors. However, this approach is fundamentally limited by training stochasticity: without access to the suspect model's exact hyperparameters (e.g., random seeds, batch ordering), behavioral divergence is inevitable even with identical data. To eliminate this confounding factor, we shift from unstable model-level comparison to differential comparison between the target dataset and non-training baselines. The core intuition is that a model behaves fundamentally differently on data it has memorized versus data it has never seen. By isolating this intrinsic behavioral gap, we can detect dataset usage independent of the model training randomness.

**Interpretability.** Even with a valid non-training baseline, auditing requires metrics that establish a causal link to dataset usage. Existing methods often rely on empirical signals like accuracy or confidence gaps. However, these are insufficient proofs, as high empirical performance can arise merely from strong generalization rather than specific data memorization. To resolve this ambiguity, DataGuard adopts an information forensics approach grounded in the Information Plane (Shwartz-Ziv & Tishby, 2017). Rather than relying on heuristic correlations, we theoretically quantify the information a model retains about inputs and labels. If the suspect model retains significantly more information from the target dataset than from non-training baselines, this asymmetry serves as statistical evidence of prior exposure, distinguishing memorization from generalization.

**Auditability.** Finally, translating auditing metrics into a rigorous auditing result requires robust hypothesis testing. Standard approaches often assume these metrics follow a known distribution (e.g., Gaussian) to perform parametric tests. However, metrics derived from complex, non-linear deep networks rarely exhibit verifiable normality, rendering such tests fragile. DataGuard circumvents this limitation by employing Hotelling's $T^2$ statistic. This multivariate formulation jointly models the variability and correlations of information metrics and, crucially, converges asymptotically to a known $\chi^2$ distribution without requiring a specific parametric distribution for the underlying metrics. This property allows us to derive quantitative auditing results without relying on unrealistic distributional assumptions.

## 3. DataGuard

### 3.1. Problem Statement

**Threat Model.** We consider a dataset auditing scenario involving a *Dataset Owner* and a *Model Provider*. The Dataset Owner holds a protected target dataset $\mathcal{D}_t = \{X_t, Y_t\}$ and suspects it was used without authorization to train a suspect model $f_s(\cdot)$. The Model Provider trained $f_s(\cdot)$ on a suspect dataset $\mathcal{D}_s = \{X_s, Y_s\}$. We consider the black-box setting, where the dataset owner has no knowledge about training configurations of $f_s(\cdot)$ and can only query it to obtain predicted probability vectors via its model API.

To perform auditing with non-training baselines, we consider a practical scenario where the dataset owner can access additional data beyond the released dataset and use it to form *auxiliary datasets* $\mathcal{D}_a$. These auxiliary datasets are drawn from the same underlying distribution as the target dataset $\mathcal{D}_t$, while being guaranteed to be non-training data. This scenario is realistic in many real-world pipelines: released datasets are typically curated from a substantially larger raw data pool through selection, filtering, and pre-processing, leaving unused portions that naturally serve as non-training baselines. For example, CIFAR-10/100 are labeled subsets curated from the 80 Million Tiny Images collection. Accordingly, we leverage these unused portions to construct a set of auxiliary datasets for auditing. When such raw pools are unavailable, or when the available auxiliary data exhibit distribution shift, we discuss the implications and practical considerations in Section 5.1.

**Auditing Goals.** Formally, DataGuard is designed to assist a dataset owner in determining whether the training dataset of a suspect model, $\mathcal{D}_s$, includes samples from a protected target dataset $\mathcal{D}_t$. Formally, we define DataGuard as a statistical function, denoted by $\mathcal{F}(\cdot)$,

$$\mathcal{F}\big(f_s(\cdot), \mathcal{D}_t\big) \in \mathbb{R} \,,$$

which produces a quantitative score reflecting the likelihood that the training dataset of $f_s(\cdot)$ includes samples from $\mathcal{D}_t$. As such, instead of yielding a simple binary decision, DataGuard aims to provide quantitative evidence to inform subsequent legal, contractual, or policy actions.

### 3.2. Preliminaries: Information Plane

DataGuard grounds its auditing metrics in the Information Plane (Shwartz-Ziv & Tishby, 2017), utilizing Mutual Information (MI) to characterize learning dynamics.

Consider a neural network parameterized by $\theta$, denoted as $f(\cdot; \theta)$. For a set of input samples $X$, their model outputs, denoted as $T = f(X; \theta)$, can be regarded as the learned representations derived from these inputs. Then, the mutual information $I(X; T)$ measures the quantity of information that the learned representations contain about the input data, quantifying how much information the model retains from the input data. Furthermore, let the ground-truth labels of $X$ be denoted by $Y$. The mutual information $I(T; Y)$ measures the information that the learned representations contain about the labels, quantifying how much task-relevant

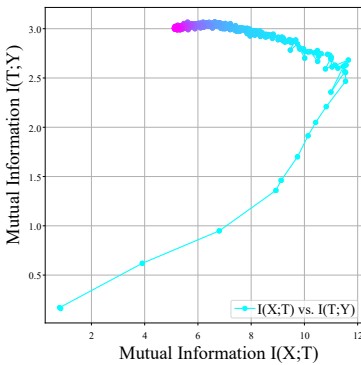

*Figure 1.* Information Plane of VGG16 on CIFAR-10. The distinct "Fitting" and "Compression" phases create a unique footprint for training data.

information the model has acquired.

Building on these metrics, the Information Plane (Shwartz-Ziv & Tishby, 2017) visualizes the training dynamics by plotting the trajectory of $(I(X;T), I(T;Y))$ pairs over epochs. Figure 1 exhibits an example of the Information Plane for a VGG-16 model trained on CIFAR-10 dataset, where each node along the trajectory represents a $(I(X;T), I(T;Y))$ pair at each training epoch, with color transitioning from cyan to purple to indicate epoch progression. Through the information plane, two distinct phases. In the initial *fitting phase*, both $I(X;T)$ and $I(T;Y)$ increase concurrently, reflecting the model memorizing dataset-specific information while learning associations with the ground truth labels at the same time. As training continues, the model enters the compression phase, in which $I(X;T)$ gradually decreases while $I(T;Y)$ stabilizes. This change indicates that the model discards prediction-irrelevant information contained in the training dataset while preserving task-relevant patterns. This compression phase spontaneously enhances the model's generalization capability on unseen data. Crucially, this training dynamic creates a unique information-theoretic footprint for training data.

### 3.3. Differential Information Forensics

The dynamics observed in the information plane reveal an important pattern to trace the dataset usage: the training dataset tends to exhibit lower $I(X;T)$ while achieving higher $I(T;Y)$ compared to the non-training dataset. To formalize this intuition, we present the following theorems to establish inequality relations for these two mutual information metrics.

First, for the mutual information between model output and the label $I(T;Y)$, we have the following theorem.

**Theorem 3.1.** *Assume that the suspect model is trained by minimizing the cross-entropy loss between the model prediction and the ground truth label, and that the training dataset $X_{train}$ and non-training data $X_{ntrain}$ are indepen-*

*dent samples drawn from the same data distribution (i.i.d). Then, we have*

$$I(T_{X_{train}}; Y_{train}) \geq I(T_{X_{ntrain}}; Y_{ntrain}) . \quad (1)$$

We provide the proof in Appendix A.1.

Second, for the mutual information $I(X;T)$, we begin by presenting the following Lemma to demonstrate its trend during the compression phase. This Lemma builds on the idea of modeling the optimization process under SGD during the compression phase as a diffusion process in statistical physics (Shwartz-Ziv & Tishby, 2017).

**Lemma 3.2.** *Consider a model as a stochastic encoder $T = f_\Theta(X)$ with training dataset $\mathcal{D}$, where the weights $\Theta$ evolve under a diffusion process that admits a Fokker–Planck description. Suppose the diffusion is run under a training constraint that fixes the empirical error (or equivalently, the label information) on the dataset $\mathcal{D}$. Let $p_\tau(\theta)$ denote the distribution of weights at time $\tau$. As $\tau \to \infty$, this process converges to a stationary distribution $p_\infty(\theta)$ that maximizes the entropy $h(\Theta)$ subject to the training constraint. Then, along the diffusion, the mutual information $I(X;T)$ for the training dataset is asymptotically non-increasing, and at stationarity, it satisfies*

$$I_{\mathcal{D}}(X; T_\infty) \leq I_{\mathcal{D}}(X; T_0) ,$$

where $T_0$ denotes the state at the beginning of the diffusion ($\tau = 0$), rather than the initial state of the entire training process. The proof details are deferred to Appendix A.2.

Consequently, the mutual information between the model output and the data sample, $I(X;T)$, decreases during the compression phase. Building on this nature, we formalize the following theorem.

**Theorem 3.3.** *Assume that the training dataset $X_{train}$ and any non-training dataset $X_{ntrain}$ are independent samples drawn from the same data distribution (i.i.d.), the mutual information between inputs and representations satisfies*

$$I(X_{train}; T_{X_{train}}) \leq I(X_{ntrain}; T_{X_{ntrain}}) . \quad (2)$$

We provide the proof in Appendix A.3. The main idea is that, during training, the diffusion dynamics tend to increase $H(T|X)$ only on the training dataset. For non-training data, no such constraint applies, so its $I(X;T)$ is not guaranteed to decrease and can remain larger than on the training set.

Theorem 3.1 and 3.3 establish a formal distinction between training and non-training datasets, providing interpretable evidence of dataset usage. In our auditing framework, the auxiliary dataset serves as the non-training baseline. Hence, we can compare the information plane metrics between the target dataset and the auxiliary dataset to assess whether the

target dataset follows consistent or different patterns of a non-training dataset. Since the auxiliary dataset is curated by the dataset owner and deliberately selected from the same raw data source, its distribution naturally matches that of the target dataset. Thus, the i.i.d. condition in the two theorems is not merely a theoretical assumption but a practical design choice. Moreover, we further discuss the impact of the non-i.i.d. condition in Section 5.1.

### 3.4. Auditing Score Calculation

Finally, we transform the observed differences in mutual information metrics between the target dataset and the auxiliary dataset into a statistically defensible auditing score. This requires a principled hypothesis testing framework that evaluates the extent of the discrepancies. To this end, DataGuard employs a Hotelling's $T^2$-based multivariate test (Hotelling et al., 1931; Anderson & Anderson, 1958) to provide a formal formulation for hypothesis testing.

#### 3.4.1. HOTELLING'S $T^2$ STATISTIC CONSTRUCTION

Formally, consider the suspect model $f_s(\cdot)$, the target dataset $\mathcal{D}_t = \{(X_t, Y_t)\}$, and a collection of auxiliary datasets $\{\mathcal{D}_a^{(1)}, \ldots, \mathcal{D}_a^{(k)}\}$, where each $\mathcal{D}_a^{(i)} = \{X_a^{(i)}, Y_a^{(i)}\}$ serves as a non-training baseline. For each dataset, we obtain the model predictions $T_t = f_s(X_t)$ and $T_a^{(i)} = f_s(X_a^{(i)})$. From these, two mutual information metrics, $I(X;T)$ and $I(T;Y)$, are computed to form a two-dimensional vector:

$$\mathbf{m}_t = \left(I(X_t;T_t),\ I(T_t;Y_t)\right)^\top, \tag{3}$$

$$\mathbf{m}_a^{(i)} = \left(I(X_a^{(i)};T_a^{(i)}),\ I(T_a^{(i)};Y_a^{(i)})\right)^\top. \tag{4}$$

The detailed calculations of mutual information are shown in Appendix B.4. Then, collecting all auxiliary metric vectors $\{\mathbf{m}_a^{(i)}\}_{i=1}^k$ yields an empirical distribution that characterizes the variability of non-training baselines.

Let $\hat{\mu}_a$ and $\hat{\Sigma}$ denote the empirical mean and covariance of the auxiliary vectors, respectively, yielding:

$$\hat{\mu}_a = \frac{1}{k}\sum_{i=1}^k \mathbf{m}_a^{(i)},\ \hat{\Sigma} = \frac{1}{k-1}\sum_{i=1}^k \left(\mathbf{m}_a^{(i)} - \hat{\mu}_a\right)\left(\mathbf{m}_a^{(i)} - \hat{\mu}_a\right)^\top.$$

Our Hotelling's $T^2$ statistic for the target dataset, denoted as $T^2_{target}$, is then defined as

$$T^2_{target} = \frac{k}{k+1} \cdot (\mathbf{m}_t - \hat{\mu}_a)^\top \hat{\Sigma}^{-1}(\mathbf{m}_t - \hat{\mu}_a), \tag{5}$$

where $k$ is the number of auxiliary datasets and the scaling factor $\frac{k}{k+1}$ accounting for the additional variance contributed by the target being a single-sample group. This formulation corresponds to a two-sample Hotelling's $T^2$

test, specialized to the case where the target group contains a single sample.

This statistic measures the extent to which the target dataset's information plane metrics deviate from the empirical distribution formed by auxiliary baselines. Recall that the two metrics, $I(X;T)$ and $I(T;Y)$, evolve in a coupled manner during training. Consequently, auditing based on either metric alone would neglect this correlation and risk biased conclusions. Hotelling's $T^2$ resolves this issue by treating the two metrics as a joint vector and normalizing their differences with respect to the covariance of auxiliary baselines. Therefore, it evaluates deviations not only in absolute magnitude but also relative to the typical co-variation patterns of auxiliary datasets, thereby offering a rigorous multivariate characterization of dataset usage.

#### 3.4.2. HYPOTHESIS TESTING

Having constructed Hotelling's $T^2$ statistic, we can now leverage it as the foundation for a formal hypothesis testing framework. Recall that the goal of dataset auditing is to assess whether the suspect model $f_s(\cdot)$ has used the target dataset $\mathcal{D}_t$ in its training. To achieve this goal, we test whether the target dataset $\mathcal{D}_t$ exhibits patterns that significantly deviate from those of non-training auxiliary datasets $\{\mathcal{D}_a^{(i)}\}_{i=1}^k$. Formally, we can cast dataset auditing as the following hypothesis testing problem:

- Null hypothesis ($H_0$): the suspect model $f_s(\cdot)$ has not used the protected target dataset $\mathcal{D}_t$ during training. In this case, the information-plane metrics of $\mathcal{D}_t$ should be statistically indistinguishable from those of non-training auxiliary datasets $\{\mathcal{D}_a^{(i)}\}_{i=1}^k$.

- Alternative hypothesis ($H_1$): the suspect model $f_s(\cdot)$ has used (full or part of) the target dataset $\mathcal{D}_t$ during training. In this case, the information-plane metrics of $\mathcal{D}_t$ will exhibit systematic deviations from the auxiliary baselines.

This formulation aligns naturally with auditing objectives, where rejecting $H_0$ corresponds to asserting statistically significant evidence of dataset usage.

In our framework, Hotelling's $T^2$ statistic quantifies the magnitude of deviation between the target dataset's metrics and the auxiliary baseline distribution. Intuitively, the larger the $T^2$ value, the less likely it is that $\mathbf{m}_t$ arises from the same distribution as $\{\mathbf{m}_a^{(i)}\}_{i=1}^k$. To enable a statistical decision, we must characterize the sampling distribution of $T^2$ under $H_0$. So, we have the following theorem to demonstrate that Hotelling's $T^2$ statistic admits a natural distribution.

**Theorem 3.4.** *Under the null hypothesis $H_0$ and mild regu-*

*larity conditions, our Hotelling's $T^2$ statistic satisfies*

$$T^2_{target} \xrightarrow{d} \chi^2_d,$$

*as $k \to \infty$, where $d = 2$ is the dimension of the vector.*

The proof leverages the multivariate central limit theorem and Slutsky's theorem, and is deferred in Appendix A.4. In practice, it is not necessary for $k$ to approach infinity for the asymptotic result to be useful. As long as the number of auxiliary datasets $k$ exceeds the dimension $d = 2$, it can provide a reliable approximation to the $\chi^2_d$ distribution. This makes the test particularly practical in our setting, since the auxiliary datasets can be flexibly constructed from data sources known not to be used in training.

Based on this asymptotic distribution, we can compute the significance of the observed deviation, i.e., $p$-value, as

$$p = \begin{cases} 1, & \text{if } \exists i, I(X_t; T_t) > I(X_a^{(i)}; T_a^{(i)}) \\ & \quad \text{or } I(T_t; Y_t) < I(T_a^{(i)}; Y_a^{(i)}), \\ \Pr\left[\chi^2_2 \geq T^2_{\text{target}}\right], & \text{otherwise.} \end{cases}$$

Here, if there exists any auxiliary dataset $\mathcal{D}_a^{(i)}$ such that the inequality conditions of the information plane are violated (i.e., the target exhibits $I(X; T)$ larger than some auxiliary or $I(T; Y)$ smaller than some auxiliary), we set $p = 1$ as a safeguard against contradictory evidence. This ensures that such logically inconsistent cases are immediately treated as non-evidence of training usage. Otherwise, the $p$-value is computed from the $\chi^2$ distribution of Hotelling's $T^2$ statistic, providing a rigorous measure of statistical significance. In this framework, a small $p$-value indicates that the probability of observing such a large deviation under $H_0$ is very low, thereby providing statistical evidence to reject $H_0$ and conclude that $\mathcal{D}_t$ has likely been used in training the suspect model $f_s$. Conversely, a large $p$-value suggests that the target dataset's metrics are consistent with the auxiliary distribution, supporting the interpretation that $\mathcal{D}_t$ was not used in training.

Given the principled statistical meaning of $p$-value, we take it as the final auditing score, i.e.,

$$\mathcal{F}\big(f_s(\cdot), \mathcal{D}_t\big) = p. \tag{6}$$

This score translates geometric deviations in the information plane into principled statistical evidence, enabling dataset owners to make defensible and interpretable judgments about dataset usage. In statistical practice, significance thresholds such as $\alpha = 0.05$ or $\alpha = 0.01$ are typically used to compare against the observed $p$-value, providing a conventional rule for deciding whether to reject the null hypothesis. Notably, these thresholds are not empirical choices, but conventional standards in hypothesis testing.

## 4. Performance Evaluations

In this section, we conduct experiments on CIFAR-10, CIFAR-100, and tiny-ImageNet with VGG, ResNet, and DeiT to evaluate the performance of DataGuard. The code for reproducing main experiments is available at https://anonymous.4open.science/r/DataGuard_anonymous/

### 4.1. Experiment Settings

The naming rule for suspect models follows the format of "structure-dataset" (e.g., "RN-CF10" refers to ResNet-18 trained on CIFAR-10). Each dataset is divided into 100 segments, with the original class distributions preserved within each segment to minimize the potential impact of segmentation on auditing performance. Across different experimental scenarios, specific segments are designated as auxiliary, target, or suspect datasets to control the overlap between suspect and target datasets. The exact assignment of segments will be specified at the beginning of each experiment. Comprehensive details regarding training configurations and data augmentation are provided in Appendix B.

### 4.2. Overall Auditing Performance

We evaluate DataGuard under two scenarios: 1) **Positive Case:** The suspect model is trained on the target dataset (segments 1–20); and 2) **Negative Case:** The suspect model is trained on disjoint data (segments 21–40), where the target dataset comprises segments 1–20. In both settings, remaining segments 41–100 form the auxiliary datasets.

***Detection Performance.*** Table 1 presents results for the positive case. DataGuard consistently yields p-values below machine precision ($< 10^{-308}$) across all architectures (VGG, ResNet, DeiT). These vanishingly small p-values confirm a statistically significant separation between target and auxiliary distributions, rejecting the null hypothesis with near-certainty. (See Fig. 3a in Appendix C for the visual distribution of the statistic). Crucially, these results hold despite extensive data augmentations (flipping, jittering), demonstrating that DataGuard captures fundamental information-theoretic footprints robust to standard generalization techniques. Conversely, for the negative case (Table 2), DataGuard consistently returns p-values of 1.0, indicating that without training exposure, the target dataset's information metrics fall within the auxiliary distribution.

***Stability Analysis.*** To assess robustness against training randomness (e.g., initialization, seed), we repeated the experiments 100 times. As detailed in Appendix C, DataGuard demonstrates exceptional stability: across all 100 runs, the False Detection Rate (FDR) remains strictly 0%, and the Correct Detection Rate (CDR) exceeds 92% (reaching 100% on CIFAR-10/100). This confirms that DataGuard delivers

*Table 1.* Dataset auditing results for the suspect model which is trained on the target dataset under different model and datasets

| | I(X;T) | | | | I(T;Y) | | | | p-value |
|---|---|---|---|---|---|---|---|---|---|
| | Target Dataset | Auxiliary Datasets | | | Target Dataset | Auxiliary Datasets | | | |
| | | min | max | mean | | min | max | mean | |
| VG-CF10 | 3.61277 | 5.46178 | 5.70761 | 5.58765 | 3.30718 | 2.84399 | 2.92389 | 2.87705 | $< 10^{-308}$ |
| VG-CF100 | 3.80883 | 5.12271 | 5.41398 | 5.28837 | 3.29040 | 2.82042 | 2.88676 | 2.85923 | $< 10^{-308}$ |
| RN-CF10 | 4.16720 | 5.47126 | 5.89560 | 5.70322 | 3.27201 | 2.58734 | 2.82029 | 2.71917 | $< 10^{-308}$ |
| RN-CF100 | 4.19711 | 5.56568 | 5.85146 | 5.68289 | 3.23220 | 2.68006 | 2.82003 | 2.76115 | $< 10^{-308}$ |
| DeiT-CF10 | 6.01209 | 6.82767 | 7.01116 | 6.93422 | 2.07523 | 1.14808 | 1.21523 | 1.17943 | $< 10^{-308}$ |
| DeiT-CF100 | 11.15009 | 11.39942 | 11.44672 | 11.42042 | 3.07942 | 2.838447 | 2.866973 | 2.85563 | $< 10^{-308}$ |

*Table 2.* Dataset auditing results for the suspect model which is not trained on the target dataset under different models and datasets

| | I(X;T) | | | | I(T;Y) | | | | p-value |
|---|---|---|---|---|---|---|---|---|---|
| | Target Dataset | Auxiliary Datasets | | | Target Dataset | Auxiliary Datasets | | | |
| | | min | max | mean | | min | max | mean | |
| VG-CF10 | 5.32696 | 5.46838 | 5.66693 | 5.55714 | 2.86879 | 2.84241 | 2.90206 | 2.86045 | 1.0 |
| VG-CF100 | 5.39220 | 5.37755 | 5.60026 | 5.48520 | 2.87498 | 2.86817 | 2.94811 | 2.90693 | 1.0 |
| RN-CF10 | 5.62152 | 5.60373 | 5.84936 | 5.73975 | 2.75178 | 2.75209 | 2.83801 | 2.79031 | 1.0 |
| RN-CF100 | 5.63205 | 5.50855 | 5.74231 | 5.66469 | 2.73313 | 2.74604 | 2.85946 | 2.81363 | 1.0 |
| DeiT-CF10 | 7.01209 | 6.82767 | 7.01216 | 6.93422 | 1.07523 | 1.14808 | 1.21523 | 1.17943 | 1.0 |
| DeiT-CF100 | 7.67688 | 7.42142 | 7.97688 | 7.70467 | 2.13728 | 1.53728 | 2.43728 | 2.00124 | 1.0 |

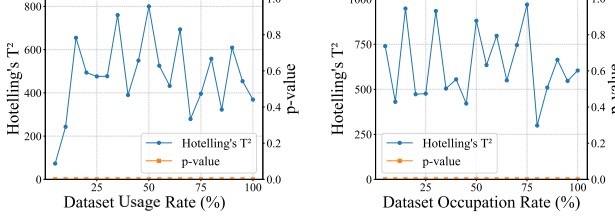

*(a)* Hotelling $T^2$ and p-value.    *(b)* Hotelling $T^2$ and p-value.

*Figure 2.* The variation of auditing scores under partial exposure.

reliable auditing performance, consistently avoiding false positives under stochastic training variations. More Results and detailed settings are provided in Appendix C.

### 4.3. Auditing under Partial Exposure

Practical auditing often involves partial exposure rather than binary inclusion. We evaluate DataGuard under two such regimes: *Sensitivity*, where the suspect model utilizes only a subset of the target dataset; and *Robustness*, where the target dataset constitutes a minor fraction of a larger training dataset. Here, we take ResNet-CF100 as an example with detailed settings and more results in Appendix E and F.

**Sensitivity.** We vary the proportion of the target dataset used by the suspect model from 5% to 100%. As shown in Figure 2a, even at a 5% usage rate, DataGuard detects a statistically significant separation between target and auxiliary metrics, with a large Hotelling's $T^2$ statistic and small $p$-values. The information footprint of the used subset remains distinct, confirming that DataGuard effectively counters evasion based on selective sample usage.

**Robustness.** We simulate a mixed-training scenario where

the target dataset comprises only 5% to 100% of the suspect model's total training data. As illustrated in Figure 2b, the information metrics of the target dataset remain consistently distinguishable from auxiliary baselines regardless of the mixing ratio. Even when the target data is diluted to a 5% occupation rate, the Hotelling's $T^2$ statistic remains large with vanishingly small $p$-values. This validates DataGuard's applicability to large-scale training scenarios where proprietary data forms only a small fraction of the corpus.

### 4.4. Comparison Results

We compare DataGuard against nine intrusive baselines, including eight representative watermarking methods (e.g., BadNets (Gu et al., 2019), Blended (Chen et al., 2017), WaNet (Nguyen & Tran, 2021), UBW-P (Li et al., 2022), UBW-C (Li et al., 2022), Label-Consistent (Turner et al., 2019), Sleeper Agent (Souri et al., 2022), and Domain Watermarking (Guo et al., 2023)). Besides, we also implement the latest membership inference-based work (Huang et al., 2024), abbreviated as MemAud. We evaluate both *auditing efficacy* (Attack Success Rate, ASR) and *collateral damage* (Degradation of Benign Accuracy, DBA; and Harmful Degree, H), follows setting proposed in (Guo et al., 2023). The detailed settings can refer to Appendix G.

*Results.* As summarized in Table 3, all methods successfully detect dataset usage while significant distinctions arise in practical utility. As shown in the DBA columns, watermarking methods and MemAud will inevitably incur a performance degradation on the model trained on the watermarked dataset. That is, they achieve dataset auditing by sacrificing the model performance. Furthermore, the Harmful Degree (H) metrics reveal deviations in predictions for

*Table 3.* The results of DataGuard and existing watermarking methods for three positive suspect models on CIFAR-10

| | RN-CF10 | | | | VG-CF10 | | | | DeiT-CF10 | | | |
|---|---|---|---|---|---|---|---|---|---|---|---|---|
| | ASR(%) | DBA(%) | H(%) | Judgment | ASR(%) | DBA(%) | H(%) | Judgment | ASR(%) | DBA(%) | H(%) | Judgment |
| BadNets | 100.00 | 1.01 | 100 | True | 100.00 | 1.13 | 100 | True | 100.00 | 0.91 | 100 | True |
| Blended | 100.00 | 0.92 | 100 | True | 99.95 | 1.52 | 100 | True | 99.82 | 1.07 | 100 | True |
| WaNet | 97.31 | 2.05 | 97 | True | 96.28 | 1.85 | 98 | True | 97.15 | 2.97 | 100 | True |
| Label-Consistent | 96.05 | 9.59 | 100 | True | 99.82 | 5.45 | 99 | True | 95.31 | 7.81 | 100 | True |
| Sleeper Agent | 94.54 | 6.47 | 94 | True | 92.30 | 2.94 | 94 | True | 94.30 | 6.02 | 98 | True |
| UBW-C | 89.21 | 5.54 | 89 | True | 88.21 | 6.01 | 90 | True | 91.25 | 3.93 | 89 | True |
| UBW-P | 92.32 | 1.94 | 86 | True | 88.67 | 1.95 | 85 | True | 88.04 | 1.22 | 90 | True |
| Domain Watermarking | 90.45 | 1.67 | 10 | True | 91.00 | 1.87 | 11 | True | 89.37 | 1.97 | 13 | True |
| MemAud | - | 1.05 | - | True | - | 1.92 | - | True | - | 0.79 | - | True |
| DataGuard (Ours) | - | 0.00 | - | $p < 10^{-308}$ | - | 0.00 | - | $p < 10^{-308}$ | - | 0.00 | - | $p < 10^{-308}$ |

watermarked data from their correct labels. Even the least intrusive method, i.e., Domain Watermarking, exhibits an H value exceeding 10%, i.e., over 10% of data with watermarks will incur the wrong predictions. In practice, dataset owners who inject watermarks should disclose the inevitable utility loss and security risks when releasing datasets for ethical considerations. Once attackers are aware of the watermarking details, they can exploit them to mislead models into misclassifying queries with similar triggers. In this way, watermarks originally designed for verification can instead become tools for adversaries. In stark contrast, DataGuard, which eschews any form of backdoor or watermark and perturbation injection, incurs zero DBA or H, underscoring its non-invasive nature and practical harmlessness.

## 5. Discussions

### 5.1. Auxiliary Dataset Assumptions

DataGuard requires access to representative *non-training baselines*, and auxiliary datasets drawn from unused portions of a raw data pool are one practical instantiation of such baselines. While our theoretical derivations assume i.i.d. auxiliary data, real-world baselines often exhibit distribution shifts. Such shifts may increase Type I error (false-detection rate) by reducing the separation between the target and non-training datasets. To assess robustness under distribution shift, we perturb class proportions of auxiliary datasets to construct non-i.i.d. non-training baselines. As reported in Appendix H, DataGuard consistently maintains high correct-detection rates and low false-detection rates, demonstrating robustness to distribution shift. These results suggest that strict distributional identity is not a hard prerequisite for DataGuard to be reliable in practice.

Moreover, in realistic settings, auxiliary data may shift due to other practical scenarios such as curation bias, time drift, label noise, or a lack of a larger reserve pool. Thus, we consider three more practical shifts as follows. For covariate shift, we constructed shifted auxiliary datasets by applying progressively stronger perturbations. The mild shift applies ColorJitter (brightness/contrast/saturation=0.1, hue =0.02); the strong setting applies heavier ColorJitter

(brightness/contrast/saturation=0.3, hue =0.08), followed by GaussianBlur (kernel size 5, $\sigma \in [0.5, 1.5]$) and RandomAffine (rotation up to $15°$). For label mismatch, we injected random label noise into the auxiliary datasets at rates of 10% and 20%. For reduced auxiliary pools, we removed 10% or 20% of auxiliary samples from the highest- and lowest-confidence tails according to the suspect model's predictions, simulating realistic filtering effects during dataset curation. The results are shown in Table 8 in Appendix H. Overall, DataGuard still remains reliable under mild mismatch, with high CDR and near-zero FDR. As the mismatch becomes stronger, FDR increases and CDR decreases, especially in more challenging settings such as DeiT/TinyImageNet. These results clarify that DataGuard is not intended to handle arbitrary auxiliary mismatch, but remains practically useful in its target regime, where reasonably representative non-training baselines can be constructed.

In certain domains, only a single fixed dataset may be available when raw data pools are inaccessible. In such cases, auxiliary-free methods and synthetic auxiliary baselines are natural extensions that we leave for future work.

### 5.2. Adversarial Robustness

Adversarial robustness is another important consideration. A recent study (Shao et al., 2025) highlighted that intrusive auditing methods are inherently unreliable under adversarial settings. Although adversarial strategies targeting non-intrusive auditing have not yet been explored, a natural question arises regarding the effect of training-time noise. To assess resilience against training-time noise, we evaluate models trained with Differential Privacy (DP-SGD, $\varepsilon = 8$). Results in Appendix I show that DataGuard retains high detection accuracy. The key insight is that although DP-SGD suppresses the absolute magnitudes of information metrics, it affects training and non-training samples symmetrically. Consequently, the relative separation between training and non-training remains intact. Moreover, we stress test DataGuard under stronger DP-SGD with smaller privacy budgets $\varepsilon$ and report the resulting CDR/FDR in Table 10 in Appendix I. These results show that DataGuard retains useful auditing signal under moderate noisy training, while the per-

formance degrades as the privacy budget becomes stricter. This behavior is consistent with the privacy-utility trade-off of DP-SGD. Stronger privacy noise reduces the model's fitting quality and compresses the distinction between training and non-training outputs, which makes auditing harder. However, the prediction performance on non-training data also deteriorates, so the separation is compressed but not completely eliminated. Consequently, the auditing signal is weakened but not immediately erased.

Nonetheless, one can envision future adversarial strategies that deliberately manipulate or obscure the information-plane structure. Investigating the feasibility of such attacks and developing corresponding defenses constitutes an important direction for future work.

### 5.3. Generalization to Foundation Models

We further evaluate DataGuard on self-supervised visual encoders (e.g., SimCLR). Experiments detailed in Appendix J demonstrate that DataGuard effectively distinguishes positive models from negative ones without false positives. This validates that the information-theoretic footprints of training data persist in self-supervised representation learning, supporting the auditing of large-scale visual foundation models. However, extending our framework to Large Language Models (LLMs) remains challenging. The primary obstacle is that textual data resides in a discrete space, rendering standard differential entropy estimators inapplicable for accurately calculating the memorization metric $I(X; T)$. Future work will focus on developing scalable MI estimators tailored for high-dimensional discrete modalities.

## 6. Conclusion

This paper proposed DataGuard, a novel non-intrusive framework for dataset auditing on machine learning models. While most existing auditing approaches rely on intrusive methods or empirical observations, DataGuard is grounded in provable information-theoretic inequalities, offering a novel direction toward more interpretable and defensible auditing. Extensive experiments validated the effectiveness of DataGuard, showcasing its accuracy, stability, sensitivity, and robustness in dataset auditing, shedding light on improving transparency in dataset usage and responsible AI system development.

## Impact Statement

This paper presents work whose goal is to advance the field of machine learning. Specifically, our proposed framework contributes to the responsible development of AI by providing a mechanism to verify dataset usage and protect intellectual property rights. This work does not involve human subjects, personal data, sensitive information, or interaction with deployed systems. All experiments are conducted on publicly available benchmark datasets and models trained locally by the authors. No vulnerabilities in deployed systems were discovered or disclosed.

## Acknowledgment

This work was supported in part by NSF under the Grants 2348452 and 2338837. Any opinions and findings expressed in the paper are those of the authors and do not necessarily reflect the views of funding agencies.

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

# Appendices

## A. Mathematical Proofs

### A.1. Proof of Theorem 3.1

*Proof.* First, based on the definition of mutual information, we have:

$$I(Y_t; T_{X_t}) = H(Y_t) - H(Y_t|T_{X_t})$$
$$I(Y_{nt}; T_{X_{nt}}) = H(Y_{nt}) - H(Y_{nt}|T_{X_{nt}})$$

Then, the conditional entropy $H(Y|T)$ can be calculated as:

$$H(Y|T) = -\mathbb{E}_{p(y,t)}\big[\log p(y|t)\big]$$
$$= -\mathbb{E}_{p(y,f(x;\theta))}\big[\log p(y|f(x;\theta))\big] \, ,$$

where $y \in Y, t \in T$. This indicates that the conditional entropy $H(Y|T)$ is the same as the cross-entropy loss between the model prediction $f(x;\theta)$ and its ground truth label $y$. Given that the suspect model is trained by minimizing the cross-entropy loss for the training data $X_t$, we have

$$H(Y_t|T_{X_t}) \leq H(Y_{nt}|T_{X_{nt}}) \, .$$

Besides, given that the training dataset $X_t$ and non-training data $X_{nt}$ are iid, we have:

$$H(Y_t) = H(Y_{nt})$$

Therefore, we have

$$\begin{aligned}
I(Y_t; T_{X_t}) &= H(Y_t) - H(Y_t|T_{X_t}) \\
&\geq H(Y_t) - H(Y_{nt}|T_{X_{nt}}) \\
&\geq H(Y_{nt}) - H(Y_{nt}|T_{X_{nt}}) \\
&\geq I(Y_{nt}; T_{X_{nt}})
\end{aligned}$$

$\square$

### A.2. Proof of Lemma 3.2

*Proof.* We begin by mathematically expressing the standard assumptions in the Lemmas:

A1 **Diffusion dynamics:** The weight process $\Theta_\tau$ satisfies the stochastic differential equation

$$d\Theta_\tau = -\nabla_\theta \mathcal{L}(\Theta_\tau)\, d\tau + \sqrt{2\beta^{-1}}\, dW_\tau,$$

where $\mathcal{L}$ is the empirical loss, $W_\tau$ is a Wiener process, and $\beta > 0$ is the inverse-noise parameter. The corresponding density $p_\tau(\theta)$ evolves by the Fokker–Planck equation under Langevin-type diffusion

$$\frac{\partial}{\partial \tau} p_\tau(\theta) = \nabla_\theta \cdot \big(p_\tau(\theta)\nabla_\theta \mathcal{L}(\theta)\big) + \beta^{-1}\Delta_\theta p_\tau(\theta),$$

which ensures ergodicity under mild convexity assumptions.

A2 **Stationarity:** As $\tau \to \infty$, the distribution converges:

$$p_\tau(\theta) \xrightarrow{\tau \to \infty} p_\infty(\theta),$$

where $p_\infty$ solves the stationary Fokker–Planck equation and maximizes the entropy

$$H(\Theta) = -\int p(\theta)\log p(\theta)\, d\theta$$

subject to the empirical risk constraint

$$\mathbb{E}_{\theta \sim p_\infty}[\mathcal{L}(\theta)] = \mathcal{L}^*,$$

for fixed empirical error $\mathcal{L}^*$.

A3 **Encoder distribution:** For each input $x$, the encoder defines a conditional distribution

$$p_\tau(t|x) = \int \delta(t - f_\theta(x))\, p_\tau(\theta)\, d\theta,$$

with the requirements

$$\forall x \in \mathcal{D}, \quad H\big(p_\tau(\cdot|x)\big) > -\infty, \quad p_\tau(t|x) \text{ is smooth in } \tau.$$

A4 **Mild regularity:** The map $\tau \mapsto h(T_\tau|X)$ is continuous, and for any two weight distributions $p_\tau, p_\tau'$, the concavity of entropy ensures

$$H(\lambda p_\tau(\cdot|x) + (1-\lambda)p_\tau'(\cdot|x)) \geq \tag{7}$$
$$\lambda H\big(p_\tau(\cdot|x)\big) + (1-\lambda)H\big(p_\tau'(\cdot|x)\big), \quad \forall \lambda \in [0,1]. \tag{8}$$

This captures the fact that mixing encoder distributions increases conditional entropy.

Given the above assumptions, we begin our proof. First, given A1, at each time $\tau$, the weights $\Theta_\tau$ are random with law $p_\tau(\theta)$. Consequently, for any input $x$, the encoder distribution takes the form

$$p_\tau(t|x) = \int \delta(t - f_\theta(x))\, p_\tau(\theta)\, d\theta,$$

that is, a mixture of deterministic encoders indexed by $\theta$. A3 ensures this conditional distribution is well defined and has finite entropy.

Then, as diffusion progresses, $p_\tau(\theta)$ spreads out. By A4, this makes each conditional distribution $p_\tau(t|x)$ more diffuse, thereby ensuring that the conditional entropy $H(T_\tau|X)$ is nondecreasing in $\tau$. However, the marginal distribution $p_\tau(t) = \int p_\tau(t|x)p(x)\, dx$ may also experience entropy growth. Here, A2 guarantees that the training constraint fixes the empirical error, or equivalently, the amount of label information preserved by the encoder. This restriction bounds the predictive capacity of $T_\tau$, preventing $H(T_\tau)$ from increasing at a rate sufficient to counterbalance the growth of $H(T_\tau|X)$. Therefore, for dataset $\mathcal{D}$, we have

$$I(X;T_\tau) = H(T_\tau) - H(T_\tau|X),$$

which is nonincreasing in $\tau$.

Finally, by A2, as $\tau \to \infty$ the distribution $p_\tau(\theta)$ converges to the stationary law $p_\infty(\theta)$, which maximizes $H(\Theta)$ under the training constraint (i.e., Gibbs distribution arising from maximum entropy with constraint on empirical loss expectation). Equivalently, this corresponds to maximizing $H(T_\infty|X)$ among all feasible encoders with fixed task performance. Therefore, the limiting encoder minimizes the mutual information:

$$I_\mathcal{D}(X;T_\infty) \leq I_\mathcal{D}(X;T_0).$$

$\square$

### A.3. Proof of Theorem 3.3

*Proof.* Consider the model as a stochastic encoder $T = f_\Theta(X)$ with weights $\Theta$ evolving under the diffusion process (e.g., SGD with additive noise). By Lemma 1, in the limit $\tau \to \infty$, the weight distribution $p_\infty(\theta)$ converges to a stationary distribution that maximizes entropy $h(\Theta)$ subject to the empirical training constraint. Equivalently, $p_\infty(\theta)$ minimizes $I(X;T)$ over the training dataset among all feasible weight distributions:

$$I_{\text{train}}(X;T_\infty) = \min_{p(\theta) \in \mathcal{F}} I_{\text{train}}(X;T),$$

where $\mathcal{F}$ is the feasible set induced by the training constraint.

Now consider evaluating the same encoder $T_\infty$ on an independent non-training dataset. Since the diffusion process was optimized only with respect to $\mathcal{D}_{train}$, it imposes no constraint on maximizing $H(T \mid X)$ outside it. Therefore, for $\mathcal{D}_{ntrain}$ the conditional entropy $H(T \mid X)$ is not guaranteed to increase during training, which yields

$$I_{\text{ntrain}}(X;T_\infty) \geq I_{\text{train}}(X;T_\infty).$$

$\square$

### A.4. Proof of Theorem 3.4

*Proof.* The mild regularity conditions in the theorem statement consist of: (i) the auxiliary metric vectors $\{\mathbf{m}_a^{(i)}\}$ are i.i.d. samples with mean $\mu_a$ and finite second moments; (ii) the covariance matrix $\Sigma = \mathrm{Cov}(\mathbf{m}_a^{(i)})$ exists and is nonsingular; (iii) the target metric vector $\mathbf{m}_t$ is independent of the auxiliaries and identically distributed under $H_0$, with mean $\mu$ and covariance $\Sigma$. These conditions ensure that the multivariate CLT and consistency of the sample covariance apply, and they are standard in the analysis of Hotelling's $T^2$ and related multivariate tests (Anderson & Anderson, 1958). With these conditions, we prove this theorem in three steps.

Then, given the null hypothesis $H_0$, we can prove Theorem 3 as follows:

**Step 1 CLT for auxiliary mean.** By the multivariate central limit theorem, since $\{\mathbf{m}_a^{(i)}\}$ are i.i.d. with mean $\mu_a$ and covariance $\Sigma$ (finite second moments), we have

$$\sqrt{k}(\hat{\mu}_a - \mu) \xrightarrow{d} \mathcal{N}(0, \Sigma).$$

**Step 2 (Difference between target and auxiliary mean).** Under $H_0$, the target metric vector $\mathbf{m}_t$ is an independent sample from the same distribution as the auxiliaries $\mathbf{m}_a^{(i)}$, with mean $\mu$ and covariance $\Sigma$. Hence,

$$\sqrt{\tfrac{k}{k+1}}\,(\mathbf{m}_t - \hat{\mu}_a) = \sqrt{\tfrac{1}{k+1}}\,\sqrt{k}(\mu - \hat{\mu}_a) + \sqrt{\tfrac{k}{k+1}}\,(\mathbf{m}_t - \mu).$$

The first term converges in distribution to $\mathcal{N}(0, \Sigma/(k+1))$ by Step 1, while the second term has distribution $\mathcal{N}(0, \Sigma)$ asymptotically. Thus, the sum converges in distribution to $\mathcal{N}(0, \Sigma)$.

**Step 3 (Consistency of covariance).** By the law of large numbers in the i.i.d. setting, the sample covariance $\hat{\Sigma}$ is a consistent estimator of $\Sigma$:

$$\hat{\Sigma} \xrightarrow{p} \Sigma, \qquad \text{and hence} \qquad \hat{\Sigma}^{-1/2} \xrightarrow{p} \Sigma^{-1/2},$$

**Step 4 (Quadratic form and Slutsky's theorem).** Define the standardized vector

$$\mathbf{Z}_k := \hat{\Sigma}^{-1/2}\sqrt{\tfrac{k}{k+1}}(\mathbf{m}_t - \hat{\mu}_a).$$

By Step 2 and Step 3 with Slutsky's theorem, $\mathbf{Z}_k \xrightarrow{d} \mathcal{N}(0, I_d)$. Therefore,

$$T_{target}^2 = \mathbf{Z}_k^\top \mathbf{Z}_k = \|\mathbf{Z}_k\|_2^2 \xrightarrow{d} \chi_d^2.$$

$\square$

## B. The Detailed Settings for Experimental Datasets and Configurations

### B.1. Datasets settings.

Our experiments are conducted on three datasets: CIFAR-10 (Krizhevsky et al., 2009), CIFAR-100 (CIF), and tiny-ImageNet (Le & Yang, 2015). We first merge their training and test sets to form unified datasets with 60,000 instances for CIFAR-10 and CIFAR-100, and 120,000 for tiny-ImageNet. Each dataset is then divided into 100 segments, with the original class distributions preserved within each segment to minimize the potential impact of segmentation on auditing performance. Across different experimental scenarios, specific segments are designated as auxiliary, target, or suspect datasets to control varying levels of overlap between suspect and target data. The exact assignment of segments will be specified at the beginning of each experiment.

### B.2. Suspect Model Configurations.

We adopt three model structures, i.e., VGG-16 (Simonyan & Zisserman, 2015), ResNet-18 (He et al., 2016), and one state-of-the-art image classification model, i.e., DeiT (Dosovitskiy et al., 2020), to train models on suspect datasets. The naming rule for our trained models follows the format of "structure-dataset". Here, VG, RN, and DeiT represent VGG-16,

*Table 4.* Dataset auditing results of Top-3 predictions for the suspect model which is trained on the target dataset under different model structures and datasets

| | I(X;T) | | | | I(T;Y) | | | | p-value |
|---|---|---|---|---|---|---|---|---|---|
| | Target Dataset | Auxiliary Datasets | | | Target Dataset | Auxiliary Datasets | | | |
| | | min | max | mean | | min | max | mean | |
| VG-CF10 | 3.25149 | 4.91560 | 5.13685 | 5.02889 | 2.97646 | 2.55959 | 2.63150 | 2.59401 | $< 10^{-308}$ |
| VG-CF100 | 3.42795 | 4.61044 | 4.87258 | 4.75953 | 2.96136 | 2.53838 | 2.59808 | 2.57331 | $< 10^{-308}$ |
| RN-CF10 | 3.75048 | 4.92413 | 5.30604 | 5.13290 | 2.94481 | 2.32861 | 2.53826 | 2.44725 | $< 10^{-308}$ |
| RN-CF100 | 3.77740 | 5.00911 | 5.26631 | 5.11460 | 2.90898 | 2.41205 | 2.53803 | 2.48504 | $< 10^{-308}$ |
| DeiT-CF10 | 5.41088 | 6.14490 | 6.31004 | 6.24080 | 1.86771 | 1.03327 | 1.09371 | 1.06227 | $< 10^{-308}$ |
| DeiT-CF100 | 10.03508 | 10.25948 | 10.30205 | 10.27838 | 2.77148 | 2.55460 | 2.58028 | 2.56714 | $< 10^{-308}$ |

ResNet-18, and DeiT, respectively. Besides, we use CF10, CF100, and TIN to represent CIFAR-10, CIFAR-100, and tiny-ImageNet. For instance, "RN-CF10" refers to a model trained on the suspect dataset of CIFAR-10, using the ResNet-18 architecture. For each model, we adopt the cross-entropy to calculate the training loss and employ the SGD optimizer to update the model parameters. The number of training epochs is 200 for ResNet-18 and VGG-16, and 300 for DeiT. The learning rate is 0.01 for ResNet-18 and VGG-16, and 0.05 for DeiT. The dropout rate is 0.2 for VGG-16, and 0.3 for DeiT, and the weight decay is $1 \times e^{-5}$.

### B.3. Training Dataset Augmentation.

We applied several data augmentations on the training dataset to improve model generalization and increase the difficulty of dataset auditing. In particular, if suspect models attempt to evade auditing by training on augmented versions of the target dataset, we expect that DataGuard still offers correct auditing scores. Specifically, we use random horizontal flipping with probability 0.5, random affine transformations with a rotation range of $\pm 10°$ and translations up to $10\%$ in both directions, and color jittering with random variations in brightness $\pm 0.2$, contrast $\pm 0.2$, saturation $\pm 0.2$, and hue $\pm 0.05$.

### B.4. Information Metrics Calculation

We query the suspect model with the target and auxiliary datasets to get the model predictions for calculating the Mutual Information metrics.

To calculate the mutual information between the dataset and the model output representation, we employ the binning methods (Shwartz-Ziv & Tishby, 2017) as follows: First, for each class, we discretize continuous model outputs by binning the confidence score for each class into equal intervals between 0 and 1, making it easier to calculate joint and marginal probabilities. For example, according to 30 equal intervals, we have the bins $\{0, 0.033, \cdots, 0.967, 1\}$. Then the logits $t = [0.01, 0.02, 0.97]$ can be transferred to $[1, 1, 30]$. Second, we create a contingency table that counts the occurrences of each combination of training data and model output bins to calculate their joint distribution $P(x, t)$ or $P(t, y)$. Then, we compute the marginal distributions $P(x)$ and $P(t)$, or $P(t)$ and $P(y)$ by summing the counts over the rows and columns of the contingency table, respectively. Third, we use these joint and marginal distributions to calculate mutual information based on the definition:

$$I(X; T) = D_{KL}[p(x, t) \| p(x)p(t)] \ ;$$
$$I(T; Y) = D_{KL}[p(t, y) \| p(t)p(y)] \ .$$

Here, the number of intervals is defined as the hyperparameter, which will influence the approximation performance and we set it as 200.

While neural estimators (e.g., MINE) exist, we adopt the binning method following (Shwartz-Ziv & Tishby, 2017) for its computational efficiency and deterministic nature, which is sufficient for capturing the relative distributional shifts in our hypothesis testing framework.

## C. Additional Results for Stability Evaluations

Model training randomness such as initialization, stochastic data augmentation, and minibatch ordering, may introduce run-to-run variability, potentially affecting the auditing performance. Therefore, to evaluate the stability of DataGuard

*Table 5.* Dataset auditing results of Top-3 predictions for the suspect model which is not trained on the target dataset under different model structures and datasets

| | I(X;T) | | | | I(T;Y) | | | | p-value |
|---|---|---|---|---|---|---|---|---|---|
| | Target Dataset | Auxiliary Datasets | | | Target Dataset | Auxiliary Datasets | | | |
| | | min | max | mean | | min | max | mean | |
| VG-CF10 | 4.79426 | 4.92154 | 5.10024 | 5.00143 | 2.58191 | 2.55817 | 2.61185 | 2.57441 | 1.0 |
| VG-CF100 | 4.85298 | 4.83980 | 5.04023 | 4.93668 | 2.58748 | 2.58135 | 2.65330 | 2.61624 | 1.0 |
| RN-CF10 | 5.05937 | 5.04336 | 5.26442 | 5.16578 | 2.47660 | 2.47688 | 2.55421 | 2.51128 | 1.0 |
| RN-CF100 | 5.06885 | 4.95770 | 5.16808 | 5.09822 | 2.45982 | 2.47144 | 2.57351 | 2.53227 | 1.0 |
| DeiT-CF10 | 6.31088 | 6.14490 | 6.31100 | 6.24080 | 0.96771 | 1.03327 | 1.09371 | 1.06227 | 1.0 |
| DeiT-CF100 | 6.90919 | 6.67928 | 7.17919 | 6.93420 | 1.92355 | 1.38355 | 2.19355 | 1.80112 | 1.0 |

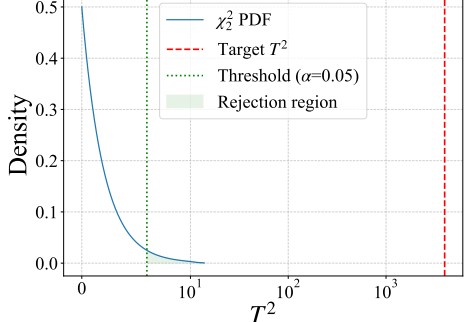

*(a)* Model trained on the target dataset.

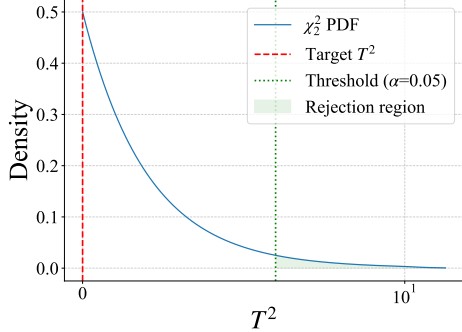

*(b)* Model does not trained on the target dataset.

*Figure 3.* The hypothesis test under $\chi_2^2$ distribution for RN-CF100.

under such randomness, we repeat each experimental setting multiple times and measure the dataset identification rate. Specifically, for each model architecture and dataset, we train 100 independent suspect models under both positive cases, where the suspect model is trained on the target dataset, and negative cases, where the suspect model is trained without any samples from the target dataset. Each model is initialized with different random seeds, affecting weight initialization, augmentation randomness, and minibatch ordering. For each trained model instance, we compute the information-plane metrics, construct Hotelling's $T^2$ statistic, and obtain the corresponding $p$-value. We then define Correct Detection Rate (CDR) and False Detection Rate (FDR) as follows:

$$\text{CDR} = \frac{\#\{\text{models in positive case with } p < \alpha\}}{100},$$

$$\text{FDR} = \frac{\#\{\text{models in negative case with } p < \alpha\}}{100},$$

where $\alpha = 0.01$ is the standard significance threshold. A higher CDR indicates that DataGuard correctly detects models that were trained on the target dataset, while a lower FDR indicates that it rarely misidentifies models that were not trained on the target dataset.

The results are shown in Table 6. Across 100 independent training runs covering all model–dataset combinations, the false-detection rate (FDR) remains consistently 0. This indicates that DataGuard does not incorrectly claim that a suspect model has used the protected dataset when it has not. For practical auditing, especially in legal, contractual, or compliance-focused environments, this "no false accusation" behavior is essential, ensuring that DataGuard does not make an unwarranted claim of dataset usage. Meanwhile, the correct detection rate (CDR) remains very high in all cases, though not always 100%. These occasional misses arise only when the suspect model itself is not well-trained (e.g., low accuracy), which weakens the expected compression effect and leads to smaller differences between training and non-training data in $I(X;T)$. This trend is reflected across datasets, where for CIFAR-10, all models achieve strong convergence, leading to a perfect 100% CDR. On the more challenging CIFAR-100, CDR drops slightly due to occasional undertrained models while on the highly complex Tiny-ImageNet, the CDR decreases further. Importantly, these reductions in CDR stem from the model's inherent training challenges rather than from limitations of DataGuard. When the suspect model reaches the standard training quality typical in practice, DataGuard consistently produces correct detections.

*Table 6.* Correct Detection Rate (CDR) and False-Detection Rate (FDR) under 100 independent training runs

| Model | CDR (%) | FDR (%) |
|---|---|---|
| VG-CF10 | 100 | 0 |
| VG-CF100 | 98 | 0 |
| VG-TIN | 92 | 0 |
| RN-CF10 | 100 | 0 |
| RN-CF100 | 100 | 0 |
| RN-TIN | 93 | 0 |
| DeiT-CF10 | 100 | 0 |
| DeiT-CF100 | 100 | 0 |
| DeiT-TIN | 94 | 0 |

Overall, these results demonstrate that DataGuard delivers stable and reliable auditing performance across all settings, consistently avoiding false positives and remaining robust under stochastic training variations.

## D. Additional Results for Top-N prediction vectors.

While many high-profile MLaaS APIs, such as Google Cloud Vision API, provide full probability vectors for debugging and interpretability purposes, some ML services restrict their output to Top-N probabilities. To evaluate DataGuard in such constrained settings, we consider the case where the model outputs only the Top-3 prediction vector. Due to the page limits, the auditing results are shown in Tables 4 and 5, which exhibit that our DataGuard still succeeds across different scenarios. This is because mutual information can still be effectively estimated from Top-$N$ ranked probabilities as long as $N > 1$, since most of the predictive information is concentrated in the highest-probability classes.

## E. Additional Results for Sensitivity Evaluations

In practice, suspect models often incorporate only a portion of the target dataset rather than using it entirely. This can occur for various reasons: developers may select specific samples to optimize model objectives, reduce computational costs, or adjust dataset composition for balance. Therefore, it is crucial for DataGuard to remain sensitive to such partial dataset usage in order to produce accurate auditing results. To evaluate this, we construct the target dataset from segments 21–40 and progressively expand the suspect dataset from segments 1–21 to 1–40, thereby increasing the proportion of target dataset usage from 5% to 100%. We compute the information plane metrics for both the target and auxiliary datasets, followed by Hotelling's $T^2$ and the corresponding $p$-value. The results are shown in Figure 4.

We observe that both $I(X;T)$ and $I(T;Y)$ are highly sensitive to the inclusion of the target dataset. Even at a usage rate as low as 5%, there is already a clear separation between the target and auxiliary datasets across both information plane metrics. Such differences are sufficient for Hotelling's $T^2$ test to yield an extremely small $p$-value, enabling DataGuard to confidently conclude that the suspect model has been trained on the target dataset. As the usage rate increases, the target dataset shares more overlap with the suspect model's training data, causing its information metrics to progressively converge toward those of the full training dataset. This further enlarges the statistical discrepancy between the target and auxiliary datasets, leading to consistently large Hotelling's $T^2$ statistics and vanishingly small $p$-values. In summary, these results validate that DataGuard remains highly effective under varying degrees of target dataset usage, providing reliable auditing capabilities even in realistic partial-usage scenarios.

## F. Additional Results for Robustness Evaluations

Mixed dataset training is a common strategy for enhancing a model's generalization capabilities, where model developers combine the target dataset with multiple other data sources to form a comprehensive training dataset. Under these circumstances, DataGuard must be robust enough to accurately audit cases where the target dataset contributes only a portion of the training data. To evaluate this, we construct suspect datasets from segments 1–21 and progressively expand the target dataset from segments 1–2 to segments 1–21, thereby increasing its proportion in the suspect dataset from 5% to 100%. We take the VGG-16 trained on CIFAR-100 as an example and compute the information plane metrics for both the target and auxiliary datasets, followed by Hotelling's $T^2$ and the corresponding $p$-value. The results are shown in Figure 5.

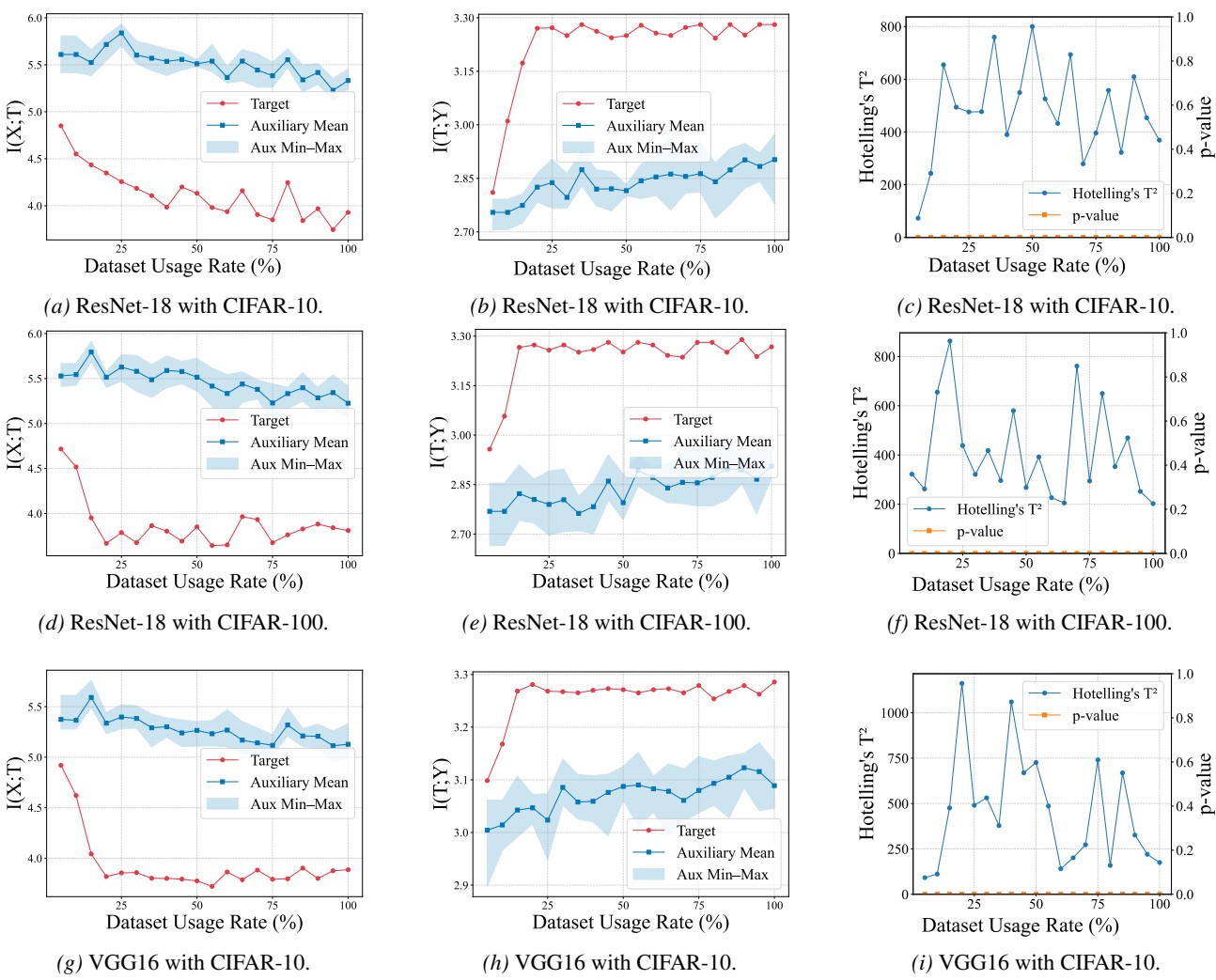

*Figure 4.* The variation of different information auditing metrics under different dataset usage rates.

The results demonstrate that DataGuard remains effective even when the target dataset contributes only a small fraction of the training data. As shown in Figure 5g, the mutual information $I(X;T)$ of the target dataset is consistently lower than that of the auxiliary datasets across all occupation rates, with a clear separation visible even at a $5\%$ occupation rate. A similar discrepancy is observed in Figure 5h for $I(T;Y)$, further reinforcing the separation between the target and auxiliary datasets. Moreover, Figure 5i exhibits that Hotelling's $T^2$ statistic remains consistently large across all occupation rates, signifying that the discrepancies between target and auxiliary datasets are statistically significant regardless of the fraction of target data involved. Correspondingly, the $p$-values remain extremely small ($< 10^{-308}$), well below any conventional significance thresholds. Together, these results demonstrate that DataGuard reliably audits dataset usage in mixed-training scenarios, underscoring its robustness.

Given the limited number of available samples, we report results down to a minimum of $5\%$ dataset occupation rate to maintain the performance of the suspect model. However, we can observe that the information metrics of the target and auxiliary datasets remain stable and exhibit a clear gap regardless of the occupation rate decrease. This arises because auxiliary datasets always resemble unseen datasets that are fundamentally different from the target dataset. Thus, even if we continue to decrease the dataset occupation rate, a distinct separation is expected, with consistently large Hotelling's $T^2$ statistics and vanishingly small $p$-values. These results suggest that large-scale data training, as in foundation models, does not hinder the effectiveness of DataGuard, and in Section 5.3 we further show that the structures and training algorithms in foundation models also pose no obstacle.

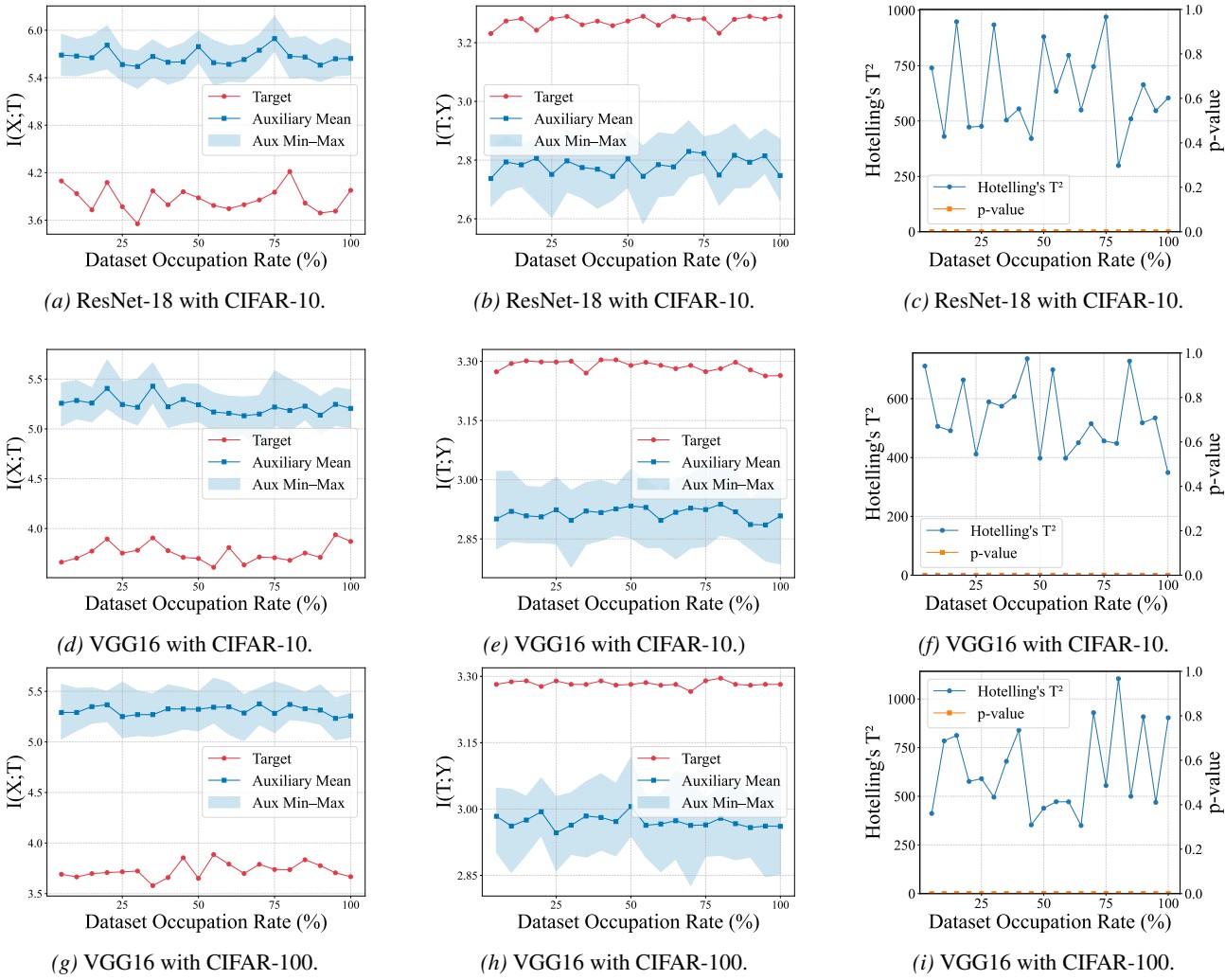

*Figure 5.* The variation of different information auditing metrics under different dataset occupation rates.

# G. Additional Results for Comparison Results

We compare our DataGuard against existing auditing methods to highlight the advantages of DataGuard.

## G.1. Counterparts

We follow the latest watermarking-based protection framework proposed in (Guo et al., 2023) and implement eight representative backdoor-based watermarking methods: BadNets (Gu et al., 2019), Blended (Chen et al., 2017), WaNet (Nguyen & Tran, 2021), UBW-P (Li et al., 2022), UBW-C (Li et al., 2022), Label-Consistent (Turner et al., 2019), Sleeper Agent (Souri et al., 2022), and Domain Watermarking (Guo et al., 2023). Besides, we also implement the latest membership inference-based work (Huang et al., 2024), abbreviated as MemAud. For the seven watermarking baselines, we conduct the backdoor injection to the targeted dataset based on the toolbox BackdoorBox (Li et al., 2023a), which integrates the implementation of these methods.

For the first seven watermarking baselines, we conduct the backdoor injection to the targeted dataset based on the toolbox BackdoorBox (Li et al., 2023a), which integrates the implementation of these methods. The watermarking rate, defined as the proportion of data samples containing the backdoor trigger, is set to 10%. The perturbation constraint, indicating the maximum allowable difference in each color channel's value, is established at 16/255. Additionally, the size of the trigger utilized in the backdoor is specified as 3×3 pixels. Besides, we implement the domain watermarking based on the method proposed in (Guo et al., 2023). For the perturbation-based baselines, we implemented the MemAud following the source code provided in (Huang et al., 2024). Then, we inject the watermark into the training dataset or generate the perturbed

*Table 7.* Correct Detection Rate (CDR) and False-Detection Rate (FDR) in 100 training runs under non-i.i.d. condition

| Model | CDR (%) | FDR (%) |
|---|---|---|
| VG-CF10 | 100 | 0 |
| VG-CF100 | 100 | 6 |
| VG-TIN | 100 | 5 |
| RN-CF10 | 100 | 0 |
| RN-CF100 | 100 | 2 |
| RN-TIN | 97 | 7 |
| DeiT-CF10 | 100 | 3 |
| DeiT-CF100 | 100 | 6 |
| DeiT-TIN | 98 | 10 |

version of the suspect dataset for training the suspect model to conduct the dataset auditing for these baselines.

### G.2. Evaluation Metrics

To evaluate the effectiveness of watermarking methods, we adopt the Attack Success Rate (ASR), defined as the proportion of verification samples (i.e., test data with embedded watermarks) classified into the designated label. A higher ASR indicates that the model trained on the watermarked dataset exhibits more distinct special predictions, making the watermarking method more effective for dataset verification. To assess the security risks associated with these baselines, we employ two metrics: Degradation of Benign Accuracy (DBA) and Harmful Degree (H). DBA measures the difference in prediction sacrifice on benign data samples (i.e., data without watermarks) from models trained on original and watermarked datasets. A larger DBA suggests a significant performance drop in models trained on the watermarked or perturbed target dataset, highlighting the adverse impact of watermarking and perturbations on model's main task accuracy. Furthermore, for the watermarking methods, the Harmful Degree (H), as defined in (Guo et al., 2023), quantifies the proportion of verification samples (i.e., testing data with watermarks) whose predicted labels deviate from their correct labels. A higher H value signifies a greater tendency of the model trained on the watermarked target dataset to misclassify watermarked or perturbed data samples, thereby indicating a more pronounced vulnerability to potential backdoor attacks.

### G.3. Comparison Results

The comparison results are listed in Table 3. It evidences that all backdoor-based watermarking methods, MemAud, and our DataGuard can successfully verify a suspect model as being trained on the targeted dataset. For the watermarking methods, the ASR consistently exceeds 88%, indicating a high rate of expected prediction for watermarked testing data, thereby confirming the models' training on the watermarked targeted dataset. MemAud, which relies on differences in membership inference accuracy, also yields the correct judgment. Our DataGuard provides an extremely low $p$-value also confirms that the suspect model was trained on the target dataset. These results underscore that DataGuard can achieve correct auditing results comparable to existing methods, but without intrusive dataset modifications.

However, in terms of practical concerns, the distinction is clear. As shown in the DBA columns of Table 3, watermarking methods and MemAud will inevitably incur a performance degradation on the model trained on the watermarked dataset. Such degradations of model accuracy range from $0.92\%$ to $9.59\%$, $1.13\%$ to $6.01\%$, and $0.91\%$ to $7.81\%$ for three models, respectively. That is, they achieve dataset auditing by sacrificing the model performance, which is unacceptable in practice. Furthermore, the Harmful Degree (H) metrics in Table 3 reveal deviations in predictions for watermarked data from their correct labels. Even the least intrusive method, i.e., Domain Watermarking, exhibits an H value exceeding 10%, i.e., over 10% of data with watermarks will incur the wrong predictions. In practice, dataset owners who inject watermarks should disclose the inevitable utility loss and security risks when releasing datasets for ethical considerations. Once attackers are aware of the watermarking details, they can exploit them to mislead models into misclassifying queries with similar triggers. In this way, watermarks originally designed for verification can instead become tools for adversaries. Moreover, since MemAud relies on membership inference, it also carries the risk of being maliciously abused for data inference attacks. In stark contrast, DataGuard, which eschews any form of backdoor or watermark and perturbation injection, incurs zero DBA or H, underscoring its non-invasive nature and practical harmlessness.

*Table 8.* Correct Detection Rate (CDR) and False-Detection Rate (FDR) in 100 training runs under three more practical shifts, where A/B represents CDR/FDR

| Model | Mild Covariate Shift | Strong Covariate Shift | 10% Label Noise | 20% Label Noise | 10% Sample Removal | 20% Sample Removal |
|---|---|---|---|---|---|---|
| VG-CF10 | 100/0 | 99/5 | 97/0 | 97/0 | 100/1 | 98/0 |
| VG-CF100 | 100/0 | 96/4 | 97/1 | 95/0 | 100/0 | 97/4 |
| VG-TIN | 99/2 | 96/6 | 97/2 | 94/2 | 95/2 | 92/5 |
| RN-CF10 | 100/0 | 99/2 | 99/0 | 100/1 | 100/0 | 99/2 |
| RN-CF100 | 100/0 | 95/5 | 98/2 | 95/2 | 97/2 | 97/2 |
| RN-TIN | 99/1 | 94/7 | 95/4 | 92/0 | 96/2 | 94/5 |
| DeiT-CF10 | 100/0 | 95/9 | 100/2 | 95/3 | 99/0 | 96/6 |
| DeiT-CF100 | 100/0 | 95/8 | 97/5 | 96/4 | 98/0 | 95/5 |
| DeiT-TIN | 96/3 | 92/11 | 95/2 | 92/6 | 92/3 | 92/8 |

*Table 9.* Correct Detection Rate (CDR) and False-Detection Rate (FDR) in 100 training runs with DP-SGD $\varepsilon = 8$

| Model | CDR (%) | FDR (%) |
|---|---|---|
| VG-CF10 | 100 | 1 |
| VG-CF100 | 98 | 1 |
| VG-TIN | 95 | 4 |
| RN-CF10 | 100 | 0 |
| RN-CF100 | 98 | 3 |
| RN-TIN | 97 | 3 |
| DeiT-CF10 | 98 | 2 |
| DeiT-CF100 | 100 | 2 |
| DeiT-TIN | 98 | 5 |

## H. Additional Results for Auxiliary Datasets Shifts

First, in Theorems 1 and 2, the i.i.d. relationship between the target and auxiliary datasets is required to support the formal theoretical reasoning. While this condition can be satisfied in modern dataset preparation pipelines, in practice, DataGuard does not require a strict i.i.d. condition. The framework only relies on the auxiliary data being representative of the same domain while not being part of the suspect model's training data. To validate this, we perturb the class-wise proportions of each auxiliary dataset to create non-i.i.d. baselines with shifted class distributions. The auditing results over 100 rounds are reported in Table 7. DataGuard consistently maintains high correct-detection rates (CDR) and low false-detection rates (FDR), demonstrating robustness to moderate distribution shift. We also observe a slight increase in FDR under class-wise perturbations. This arises from the fact that distribution shifts inherently enlarge the discrepancy between the target and auxiliary datasets, thereby amplifying the statistical contrast in their information metrics. As a result, the discrepancy introduced by distribution shift may be confounded with that arising from model training, slightly increasing FDR.

Moreover, in realistic settings, auxiliary data may shift due to other practical scenarios such as curation bias, time drift, label noise, or a lack of a larger reserve pool. Here, we consider three more practical shifts as follows. For covariate shift, we constructed shifted auxiliary datasets by applying progressively stronger perturbations. The mild shift applies ColorJitter (brightness/contrast/saturation=0.1, hue =0.02); the strong setting applies heavier ColorJitter (brightness/contrast/saturation=0.3, hue =0.08), followed by GaussianBlur (kernel size 5, $\sigma \in [0.5, 1.5]$) and RandomAffine (rotation up to $15°$). For label mismatch, we injected random label noise into the auxiliary datasets at rates of 10% and 20%. For reduced auxiliary pools, we removed 10% or 20% of auxiliary samples from the highest- and lowest-confidence tails according to the suspect model's predictions, simulating realistic filtering effects during dataset curation. The auditing results over 100 rounds are shown in Table 8. Overall, DataGuard still remains reliable under mild mismatch, with high CDR and near-zero FDR. As the mismatch becomes stronger, FDR increases and CDR decreases, especially in more challenging settings such as DeiT/Tiny-ImageNet. These results clarify that DataGuard is not intended to handle arbitrary auxiliary mismatch, but remains practically useful in its target regime, where reasonably representative non-training baselines can be constructed.

*Table 10.* Correct Detection Rate (CDR) and False-Detection Rate (FDR) in 100 training runs with DP-SGD under Varying $\varepsilon$ Values

| Model | Epsilon | CDR (%) | FDR (%) |
|---|---|---|---|
| RN-CF100 | 8 | 98 | 4 |
| RN-CF100 | 4 | 95 | 7 |
| RN-CF100 | 2 | 92 | 9 |
| RN-CF100 | 1 | 89 | 10 |
| DeiT-TIN | 8 | 98 | 5 |
| DeiT-TIN | 4 | 92 | 8 |
| DeiT-TIN | 2 | 90 | 10 |
| DeiT-TIN | 1 | 86 | 13 |

## I. Additional Results for Adaptive Methods

Adversarial robustness is another important consideration. A recent study (Shao et al., 2025) highlighted that intrusive auditing methods are inherently unreliable under adversarial settings. Although adversarial strategies targeting non-intrusive auditing have not yet been explored, a natural question arises regarding the effect of training-time noise. To assess this, we conduct an additional experiment where the suspect model is trained with DP-SGD. To maintain reasonable model performance while still introducing meaningful noise, we set the privacy budget to $\varepsilon = 8$ with a noise multiplier calibrated accordingly. The auditing results, reported in Table 9 over 100 rounds, show that DataGuard still achieves high correct-detection rates (CDR) and low false-detection rates (FDR). This arises from the fact that while DP-SGD lowers the absolute values of both $I(X;T)$ and $I(T;Y)$, it affects training and non-training samples in a similar manner during inference. As a result, the relative separation between the target and auxiliary datasets remains preserved. Since DataGuard fundamentally relies on relative rather than absolute information-plane magnitudes, this preserved relative contrast remains sufficient for reliable auditing. Moreover, we stress test DataGuard under stronger DP-SGD with smaller privacy budgets $\varepsilon$ and the resulting CDR/FDR are shown in Table 10. These results show that DataGuard retains useful auditing signal under moderate noisy training, while the performance degrades as the privacy budget becomes stricter. This behavior is consistent with the privacy-utility trade-off of DP-SGD. Stronger privacy noise reduces the model's fitting quality and compresses the distinction between training and non-training outputs, which makes auditing harder. However, the prediction performance on non-training data also deteriorates, so the separation is compressed but not completely eliminated. Consequently, the auditing signal is weakened but not immediately erased.

## J. Additional Results for Auditing Visual Foundation Models

To further demonstrate the effectiveness of DataGuard in practice, we extend our evaluation to foundation model structures. In particular, we focus on visual encoders, which are widely adopted foundation models in the image domain. Visual encoders are commonly trained with self-supervised methods to learn generalized image representations from unlabeled data. Once trained, these encoders can be used as feature extractors across a range of vision tasks for diverse downstream applications.

Here, we train the visual encoders following the popular SimCLR (Chen et al., 2020) framework on CIFAR-10/100 and Tiny-ImageNet datasets. We choose ResNet-18 as the default architecture for the base encoder and adopt the NT-Xent loss to maximize the cosine similarity for each positive pair and minimize the similarity for each negative pair. Following the similar settings in Section 4, we consider two scenarios. First, the suspect model is fully trained on the target dataset: both the suspect and target datasets consist of segments 1–20, which is referred to as the positive model. Second, the suspect model is trained without any samples from the target dataset: the suspect dataset consists of segments 1–20, while the target dataset consists of segments 21–40, which is referred to as the negative model. The remaining segments 41–100 are partitioned into ten auxiliary datasets, each containing six segments.

Note that although foundation models are trained without labels, the application of DataGuard does not require the suspect model to provide labels. Dataset auditing is conducted by the dataset owner, who always has full control over the target and auxiliary datasets and therefore, can prepare or attach ground-truth labels for these datasets. These labels are not used by the suspect model during training; instead, they are only used locally by the dataset owner to compute $I(T;Y)$. Since ground truth labels are available in our experiments, we can directly compute $I(T;Y)$.

*Table 11.* Dataset auditing results of DataGuard for the SimCLR model under different model structures and datasets

| | I(X;T) | | | | I(T;Y) | | | | p-value |
|---|---|---|---|---|---|---|---|---|---|
| | Target Dataset | Auxiliary Datasets | | | Target Dataset | Auxiliary Datasets | | | |
| | | min | max | mean | | min | max | mean | |
| Positive Models | | | | | | | | | |
| SimCLR-CF10 | 4.12677 | 6.28178 | 6.77514 | 6.58715 | 4.30718 | 3.14379 | 3.72779 | 3.47752 | $< 10^{-308}$ |
| SimCLR-CF100 | 4.30876 | 6.12271 | 6.82108 | 6.48917 | 4.37001 | 3.22472 | 3.67866 | 3.49823 | $< 10^{-308}$ |
| SimCLR-TIN | 6.76715 | 7.71725 | 8.80521 | 8.20317 | 5.95601 | 4.37734 | 5.02029 | 4.71778 | $< 10^{-308}$ |
| Negative Models | | | | | | | | | |
| SimCLR-CF10 | 6.37976 | 6.12838 | 6.70693 | 6.45714 | 3.61879 | 3.28441 | 3.90061 | 3.50145 | 1.0 |
| SimCLR-CF100 | 6.69120 | 6.10755 | 6.77261 | 6.48701 | 3.49782 | 3.26417 | 3.68214 | 3.40837 | 1.0 |
| SimCLR-TIN | 7.91752 | 7.63793 | 8.84961 | 8.17757 | 4.78718 | 4.20915 | 4.93801 | 4.50012 | 1.0 |

The performance of DataGuard on visual encoders is shown in Table 11. DataGuard consistently yields a reasonable auditing score for both positive and negative models, demonstrating its robustness and reliability in accurately auditing visual encoders. For the positive models, where the suspect models are indeed trained on the target datasets, DataGuard always reports extremely small $p$-values. This outcome indicates that the target dataset exhibits information patterns ($I(X;T)$ and $I(T;Y)$) that are statistically distinguishable from those of the auxiliary datasets. Such discrepancies faithfully reflect the role of the target dataset as training data, confirming DataGuard's ability to detect its usage even within visual encoder structures. For the negative models, DataGuard correctly identifies no evidence of training on the target dataset. In these cases, the mutual information values of the target dataset consistently fall within the range observed across the auxiliary datasets, showing strong alignment between them. Consequently, DataGuard outputs a $p$-value of 1, signifying that no statistically significant difference is detected. This result confirms that the suspect models were not trained on the target datasets, and importantly, no false positives are observed.

DataGuard's effectiveness in auditing visual foundation models is particularly significant, as these models are trained on extensive datasets that are more likely to contain copyrighted or sensitive material and are increasingly integrated into sensitive domains, such as healthcare and finance. By enabling accurate dataset auditing, DataGuard provides a concrete mechanism for ensuring data-use compliance and supporting responsible AI development. Here, we merely evaluate DataGuard on our self-trained suspect models to avoid potential ethical and legal issues, as directly auditing third-party models without authorization can raise concerns. In our future work, we plan to carefully extend DataGuard to empirical studies on real-world models.

