# OpenReview forum: "DataGuard: A Non-intrusive Dataset Auditing Framework via Differential Information Forensics"
_ICML.cc/2026/Conference — ICML 2026 regular_

### Official Review · Reviewer_GyDQ · 2026-03-11

**Soundness:** 2
**Presentation:** 3
**Significance:** 2
**Originality:** 3
**Overall Recommendation:** 3
**Confidence:** 4

**Summary:**

This paper introduces DataGuard, a novel and non-intrusive framework designed to audit whether a suspect machine learning model was trained on a specific, protected dataset. Unlike existing intrusive methods that inject watermarks which inherently degrade model performance and introduce security vulnerabilities, DataGuard preserves dataset integrity by operating without any modifications to the original data. The framework leverages differential information forensics, utilizing Mutual Information metrics derived from the Information Plane to theoretically quantify the information a model retains about its inputs and ground-truth labels. By establishing formal mathematical inequalities, DataGuard compares these metrics between the target dataset and selected non-training auxiliary datasets to identify distinct training footprints. It then applies Hotelling's $T^2$ multivariate statistical test to translate these information-theoretic discrepancies into a rigorous, quantitative auditing score.

**Compliance With Llm Reviewing Policy:**

Affirmed.

**Key Questions For Authors:**

1.	The framework requires probability vectors to compute Mutual Information, and Appendix D demonstrates success with top-3 predictions. However, many commercial MLaaS APIs only output a top-1 hard label. How does DataGuard perform under a strict top-1 constraint?
2.	To calculate $I(X;T)$, the paper relies on a deterministic binning method using 200 equal intervals. How sensitive is the Hotelling's $T^2$ statistic to the choice of this hyperparameter, particularly if the suspect model applies heavy temperature scaling or smoothing to its logits?
3.	The framework assumes the dataset owner can construct auxiliary datasets drawn from the same underlying distribution as the target dataset. In domains where data is fully exhausted or highly specialized, this can be impossible. Have the authors considered or tested whether synthetic data could serve as a viable alternative?
4.	Could the authors provide a direct empirical comparison of DataGuard against Maini et al. (2021) and Du et al. (2025) on the same CDR/FDR metrics reported in Table 6, under identical experimental conditions?

**Limitations:**

yes

**Strengths And Weaknesses:**

Strengths
•	The fusion of the Information Plane framework with Hotelling's $T^2$ statistic establishes a highly innovative non-intrusive auditing pipeline.
•	The paper provides a formal theoretical justification for its empirical observations. It establishes rigorous inequalities for $I(X;T)$ and $I(T;Y)$ to differentiate training from non-training data.
•	Unlike existing watermarking approaches, DataGuard achieves high detection rates without degrading the suspect model's benign accuracy. Furthermore, it avoids introducing security vulnerabilities.
•	The experimental design rigorously evaluates partial data exposure and mixed-training scenarios. It demonstrates high sensitivity and robust detection even at a minimal 5% usage rate.
•	The presentation is logically structured. It smoothly transitions from the problem statement to differential information forensics, and culminates in a clear statistical testing formulation.

Weaknesses
•	The framework heavily assumes the availability of auxiliary non-training datasets drawn from the same underlying distribution. This requirement is often unrealistic for proprietary (e.g., artistic works, medical images, private corpora) datasets.
•	Calculating the mutual information metric $I(X;T)$ relies on a simple binning method using 200 equal intervals. This discrete approximation is potentially sensitive to hyperparameter choices and may struggle to scale smoothly.
•	While the paper evaluates the framework using top-3 prediction vectors, real-world MLaaS APIs frequently only provide top-1 hard labels. The paper does not address how the methodology would function under these stricter constraints.
•	DP-SGD with ε = 8 is an extremely weak privacy guarantee (effectively near-zero noise in practice). The paper acknowledges that deliberate information-plane manipulation attacks haven't been studied, but provides no lower bound on what perturbation budget would break DataGuard.

---

> ### Author Rebuttal · Authors · 2026-03-30
>
> We thank the reviewer for appreciating the novelty and theoretical value of our framework. We view our theoretical component as a shift from heuristic signal-based auditing toward a theoretically grounded auditing method for a problem with potential legal implications. We agree that DataGuard does not fully resolve all practical constraints, and we have added new experiments per your comments.
>
> **Regarding Top-1 label**
>
> We agree that DataGuard is not directly applicable under a strict top-1 setting. A natural extension is to query the model multiple times on augmented versions of the same input and aggregate the returned labels into a vector, which can then be discretized to define a surrogate T. This is motivated by label-only membership inference, where training data tend to induce more stable predictions.
>
> In a preliminary experiment on CIFAR-10 with ResNet-18, using 20 augmented queries per sample over 100 repeated runs, this surrogate construction achieved a CDR of 84% and an FDR of 16%. This suggests that some auditing signal may remain available even in the label-only setting, although the current construction is not yet reliable enough to support our main claims.
> Importantly, this surrogate does not exactly match the T used in our theoretical analysis. Thus, we view top-1 auditing as an important direction and seek new theories to support and improve such methods.
>
> **Regarding MI calculation**
>
> We performed an ablation over the number of bins (50, 100, 200) and temp (0.5, 1, 2, 5). The table below reports the results on RN-CF100. The auditing performance is stable, suggesting that DataGuard is not tied to the specific 200 bins. The only clear degradation appeared at the setting of 50 bins and Temp = 5, where aggressive output smoothing and low-resolution discretization jointly suppress the information needed for MI estimation.
> We also observe that increasing the number of bins mitigates the degradation under heavy smoothing. This is consistent with the intuition that finer discretization preserves more information from softened predictive distributions, albeit at a higher computational cost. Here, using 200 bins requires about 30 mins on an RTX 5090, which we believe is still practical in the intended auditing setting.
>
> Bins|Temp|CDR|FDR
> -|-|-|-|
> 50|0.5|100|0
> 50|1|100|0
> 50|2|98|4
> 50|5|81|15
> 100| 0.5|100|0
> 100|1|100|0
> 100 |2|100|0
> 100| 5|97|0
> 200| 0.5|100|0
> 200|1|100|0
> 200| 2|100|0
> 200| 5|100|0
>
> **Regarding synthetic auxiliary datasets:**
>
> We agree that auxiliary datasets are a practical limitation of DataGuard, especially in data-exhausted domains. However, the synthetic data is also subtle: in such scarce-data settings, training a generative model to produce suitable auxiliary data may itself be difficult. Thus, we studied a conservative proxy by constructing auxiliary datasets from strongly augmented versions of the protected dataset. The results are as follows:
>
> Model|CDR|FDR
> -|-|-
> VG-CF10| 94|22
> VG-CF100| 96|24
> VG-TIN| 92|29
> RN-CF10| 98| 30
> RN-CF100| 95| 32
> RN-TIN|92|40
> DeiT-CF10|95|35
> DeiT-CF100| 92|36
> DeiT-TIN| 89|41
>
> We can see that CDR remains high, which is intuitive because when the suspect model was trained on the protected dataset, the protected dataset remains close to its augmentation-based variants. However, FDR increases substantially. This indicates that augmentations do not adequately capture the MI distribution of general non-training datasets, so even when the protected dataset was not used for training, its MI may still differ from those of the auxiliaries, leading to more false positives. These results suggest that augmentation-derived proxies do not provide reliable calibration by default. By extension, synthetic auxiliaries would also require careful validation.
>
> **Regarding comparison to dataset inference baselines:**
>
> Among the dataset inference methods in Du et al. (2025), the directly comparable baseline is Maini et al. (2021). By contrast, *Dataset Inference for Self-Supervised Models* is designed for self-supervised encoders and is not aligned with our setting. Under the same CDR/FDR evaluation setting, the results are:
>
>  Model|CDR|FDR
> -|-|-
> VG-CF10|95|8
> VG-CF100|94|7
> VG-TIN|90|12
> RN-CF10| 95|7
> RN-CF100| 95|8
> RN-TIN|91|9
> DeiT-CF10|92|12
> DeiT-CF100|91|13
> DeiT-TIN|87|13
>
> We observe that Maini et al. achieves a similar CDR to DataGuard, but with a noticeably higher FDR. One reason is that their method relies on confidence-regression signals against an unseen reference dataset, but does not provide an explicit training vs. non-training inequality that can be used as a conservative rejection rule in hypothesis testing. Our MI-based analysis yields such a theoretical criterion, providing an additional mechanism for conservatively rejecting ambiguous cases and thereby helping reduce false positives.
>
> We will add all these results to the final version. We hope these clarifications help address the reviewer’s concern and support a more positive overall assessment.

---

> > ### Author Rebuttal · Reviewer_GyDQ · 2026-04-02
> >
> > Thanks for the updates and rebuttal. Still, I believe there is novelty in this work. My concerns related to high \epsilon parameter stays. Hence, I keep my score.

---

> > > ### Author Response · Authors · 2026-04-03
> > >
> > > We thank the reviewer for this point. We agree that $ \epsilon=8 $ should not be interpreted as a strong privacy guarantee, so the original DP-SGD experiment should only be viewed as an initial noisy-training stress test.
> > >
> > > Due to the rebuttal word limit, we were only able to prioritize the key questions you listed. To better address this concern, we add experiments on stricter privacy budgets (smaller $\epsilon$) and report the resulting CDR/FDR as follows:
> > >
> > > | Model | Epsilon | CDR (%) | FDR (%) |
> > > |---|---:|---:|---:|
> > > | RN-CF100 | 8   | 98  | 4  |
> > > | RN-CF100 | 4   |  95 | 7  |
> > > | RN-CF100 | 2   |  92 | 9  |
> > > | RN-CF100 | 1   | 89| 10  |
> > > | DeiT-TIN | 8   | 98  |  5 |
> > > | DeiT-TIN | 4   |  92 |  8 |
> > > | DeiT-TIN | 2   |   90|  10 |
> > > | DeiT-TIN | 1   | 86|  13 |
> > >
> > > These results show that DataGuard retains useful auditing signal under moderate noisy training, while the performance degrades as the privacy budget becomes stricter. This behavior is consistent with the privacy-utility trade-off of DP-SGD. Stronger privacy noise reduces the model’s fitting quality and compresses the distinction between training and non-training outputs, which makes auditing harder. However, the prediction performance on non-training data also deteriorates, so the separation is compressed but not completely eliminated. Consequently, the auditing signal is weakened but not immediately erased. We will revise the paper to add this experiment.

---

### Official Review · Reviewer_6bSw · 2026-03-12

**Soundness:** 3
**Presentation:** 2
**Significance:** 2
**Originality:** 3
**Overall Recommendation:** 4
**Confidence:** 3

**Summary:**

This paper proposes a method to assess whether a dataset has been used or not for a classification model's training. It does not need to modify the dataset and can then work on any already published dataset. It relies on comparing how much information the model retains about a dataset vs similar non-training ones, through mutual information metrics, not simple accuracy. If the model retains significantly more information about a dataset over a similar one (lower $MI(X;f(X,\theta)$) due to information compression, higher $MI(f(X,\theta);Y)$ due to cross-entropy minimization), we can deduce that it has been trained on it. This mutual information grounding allows mathematical guarantees. The method can be applied to any classifier model but the mathematical guarantees only apply for supervised classifiers trained with cross-entropy and SGD-like diffusion dynamics. The method requires black-box access to the model, with full class distribution as output (not only the predicted class). The conducted experiments focused on image classifiers and therefore image datasets.

**Compliance With Llm Reviewing Policy:**

Affirmed.

**Key Questions For Authors:**

- Regarding the binning of a t vector, with e.g. 10 classes and 200 bins each, the joint space is 200^10 ≈ 10^23, which is astronomically larger than CIFAR-10's ~60,000 samples. If the distribution of T was near uniform the empirical P(t) would be essentially 1/N, N being the total number of samples — almost every sample lands in a unique bin combination, making the MI estimate degenerate. So I thought maybe it would be good to explain why this approach is still well working (for instance because very skewed distribution).

- In the following of previous commentary, could it be interesting to treat T as just the max confidence, as this could weaken the black-box accesss hypothesis to a stricter one that would only require the predicted class rather that the full distribution?

- Isn't I(X,T) = H(T) as T fully described by X? Does this could simplify some computations?

**Limitations:**

Limitations regarding the quite restricted scope of both theoretical assumptions and experiment is not reallly discussed.

**Strengths And Weaknesses:**

1. *Soundness*:

**Strengths:**
The general theoretical analysis is very clean and sound. The Hotelling’s T^2 statis-
tic seems adequate for the task. The use of mutual information is well capturing the complexity of memorization process, much richer than a raw performance metric over the dataset like accuracy.

The experimental design explores the global method performance and also two meaningful insights: how small can be the fraction of the target dataset on which the model has been trained while still being detected, and how small can be the fraction of the total training data that the target dataset represents while still being detected.

**Weaknesses:**
The theoretical guarantees apply for quite specific setup: supervised classifiers trained with cross-entropy and SGD-like diffusion dynamics.

The experiment only explores image classifiers.

The claim "However, metrics derived from complex, non-linear deep networks rarely exhibit verifiable normality, rendering such tests fragile" is missing a citation to support why these tests are considered fragile.

2. *Presentation*:

**Strengths:**
Overall quite clear and well structured paper, easy to read.

**Weaknesses:**

Some structural problems

- The abstract and introduction don't explain what types of models you are testing or what the system concretely relies on. → Classifiers (xp on image classifiers), relies on comparison of predictions on training dataset with predictions on a similar but distinct dataset.
- The core idea of distinguishability in "If the suspect model retains significantly more information from the target dataset than from non-training baselines, this asymmetry serves as statistical evidence of prior exposure, distinguishing memorization from generalization" is finally explained, but it comes too late in the paper and could be even more straightforwardly phrased.
- The threat model on page 3 mentions the user "can only query it to obtain predicted probability vectors via its model API", which is the first time (sorry if I’m wrong) we realize these are black-box classifiers. This context belongs in the introduction.

Some unclear formulations:

- The sentence "Our results demonstrate that DataGuard identifies all models trained on target dataset" lacks of information. You should add a specific qualification of the identification or just remove the sentence.
- "Moreover, DataGuard also demonstrates accurate auditing performance in cases where only a partial target dataset is used to train the suspect model and is robust enough to accurately audit instances where the target dataset constitutes only a portion of the suspect model’s training data" → It says the same thing about partial datasets twice.
- You don't need to add the phrase "by preserving model performance and mitigating security risks" in the sentence "It also outperforms existing dataset-intrusive techniques by preserving model performance and mitigating security risks".
- The section "These commitments introduce three key challenges: 1) Integrity: extracting evidence intrinsic to the training process without modifying the dataset, 2) Interpretability: employing mathematically grounded rather than heuristic signals, and Auditability: producing reliable scores that avoid false positives while remaining rigorous" is redundant and confusing because it just repeats your contributions but with different wording.
- The sensitivity and robustness could be more clearly defined I think. Like: Sensitivity, the suspect model trained on only a fraction of the target dataset; Robustness, the target dataset is only a small fraction of the suspect model's total training data.
- "For a set of input samples X, their model outputs, denoted as T = f (X; θ), can be regarded as the learned representations derived from these inputs." In this sentence, it is an unclear whether T maps to C^d or ([0,1]^#C)^d where C is the set of labels and d = #X. I understand with the threat model and the appendix B.4 that it is ([0,1]^#C)^d but I think it could be more clear.

3. *Significance*:

Since this outperforms previous techniques and that those ones require dataset modification prior to training, this work is very significant. However, the domains covered are not as broad as the literature that also proposes method for other ML models such as LLMs, LSTM or image generative models (DALL-E 3, CLIP).

4. *Originality*:

Having mathematical guarantees and no need to modify the dataset is very new to the literature.

---

> ### Author Rebuttal · Authors · 2026-03-30
>
> We thank the reviewer for appreciating the novelty and theoretical value of our framework. We also appreciate the presentation suggestions and will revise the paper accordingly in the final version, including clarifying the current scope more explicitly in the introduction and limitations.
>
> **Regarding the binning:**
>
> In principle, a naive Cartesian discretization of a C-dimensional probability vector with 200 bins per coordinate would indeed induce an astronomically large ambient grid. However, in our setting, the model outputs do not populate this grid anywhere close to uniformly. The prediction vectors lie on the probability simplex $\sum_c t_c = 1$, are highly skewed, and occupy only a small structured subset of the ambient space. In particular, most probability mass is concentrated on a few classes, while the remaining coordinates are near zero, so the effective support is far smaller than the worst-case count $200^C$ suggests.
>
> Besides, our objective is not to recover the full high-dimensional density of T with high fidelity, but to extract a consistent statistic that separates training from non-training datasets. Even if the discretized MI estimator is biased in absolute value, it can still be useful as long as it is applied consistently across target and auxiliary datasets and preserves the relative separation. The strong empirical separation observed in our experiments suggests that the estimator is not degenerating into the one sample per occupied bin regime in practice.
>
> We agree that the estimator is heuristic and may depend on the output geometry and binning choice. To address this concern, we performed an ablation over the number of bins (50, 100, 200) and temp (0.5, 1, 2, 5). The table below reports the results on RN-CF100. The auditing performance is stable, suggesting that DataGuard is not tied to the specific 200 bins. The only clear degradation appeared at the setting of 50 bins and Temp = 5, where aggressive output smoothing and low-resolution discretization jointly suppress the information needed for MI estimation.
> We also observe that increasing the number of bins mitigates the degradation under heavy smoothing. This is consistent with the intuition that finer discretization preserves more information from softened predictive distributions, albeit at a higher computational cost. Here, using 200 bins requires about 30 mins on an RTX 5090, which we believe is still practical in the intended auditing setting.
>
> Bins|Temp|CDR|FDR
> -|-|-|-|
> 50|0.5|100|0
> 50|1|100|0
> 50|2|98|4
> 50|5|81|15
> 100| 0.5|100|0
> 100|1|100|0
> 100 |2|100|0
> 100| 5|97|0
> 200| 0.5|100|0
> 200|1|100|0
> 200| 2|100|0
> 200| 5|100|0
>
> **Regarding Label-only setting:**
>
> We thank the reviewer for raising this point. We would like to clarify that using the maximum confidence and using the predicted class correspond to two different black-box settings: the former still requires access to a confidence score, while the latter is a stricter label-only setting.
>
> When only the maximum confidence is used, this lower-dimensional summary discards much of the ranking and margin information contained in the full prediction vector, so we would expect weaker separation between training and non-training datasets. In particular, under smoothed outputs or closely competing classes, the auditing signal may deteriorate. This is why we use the full prediction vector in the current formulation.
>
> On the other hand, for the strict label-only case, our current DataGuard pipeline is not directly applicable, since it relies on confidence-bearing prediction vectors to construct the information-plane statistics. A natural extension is to query the model multiple times on augmented versions of the same input and aggregate the returned labels into a vector, which can then be discretized to define a surrogate T. This is motivated by label-only membership inference, where training data tend to induce more stable predictions.
>
> In a preliminary experiment on CIFAR-10 with ResNet-18, using 20 augmented queries per sample over 100 repeated runs, this surrogate construction achieved a CDR of 84% and an FDR of 16%. This suggests that some auditing signal may remain available even in the label-only setting, although the current construction is not yet reliable enough to support our main claims. We therefore view both max-confidence and strict label-only variants as interesting future directions, rather than direct replacements for the current formulation.
>
>
> **Regarding  I(X;T) Calculation:**
>
> More generally, whenever T is a deterministic function of X, we indeed have I(X;T)=H(T), since H(T∣X)=0. Thus, once the output representation T is fixed, the computation can be simplified to estimating only the marginal distribution of the discretized T. The practical challenge then is not the identity itself, but the quality of entropy estimation under the chosen discretization, especially when T is high-dimensional.
>
> We hope these clarifications help address the reviewer’s concerns.

---

> > ### Author Rebuttal · Reviewer_6bSw · 2026-04-02
> >
> > I thank the authors for these answers that help clarify my concerns.
> >
> > Regarding binning, I'm not sure we understood each other, for instance, the experiment is not really answering my question. My question was like how come not only unique combinations, because 60k << 10^23 even with very skewed distribution. However, as I might have a mistaken representation of this, it is ok that we leave this point aside.
> >
> > Regarding label-only setting:
> > I'm surprised that a strict label-only case would not be applicable. Can't this be turned into such a vector [0,0,...0,1,0,...,0] (setting binning to 1)?
> >
> > Regarding I(X;T) Calculation:
> > I was also asking if this could simplify some of the computations (theorems/notations/proofs?).

---

> > > ### Author Response · Authors · 2026-04-03
> > >
> > > 1. Regarding binning:
> > >
> > > We appreciate the reviewer’s clarification. We now understand that the key concern is whether the discretized outputs enter an almost one-sample-per-cell regime. Our intuition is that this does not happen under reasonable binning resolutions for well-trained models, because their prediction vectors can be highly structured. Thus, many samples exhibit strongly concentrated confidence on their ground-label class, with similar values across samples, so the effective occupied support after discretization is much smaller than the ambient Cartesian grid.
> > >
> > > To better address this point, we add a small analysis on CIFAR-10 to quantify the occupancy of discretized output patterns under different binning values:
> > >
> > > | #Bins | #Samples | #Unique discretized patterns |
> > > |------:|---------:|-----------------:|
> > > | 10    | 10000    | 161                         |
> > > | 50   | 10000    | 2054                         |
> > > | 200 | 10000    | 3722                         |
> > > | 500 | 10000    | 8327                         |
> > > | 1000   | 10000    | 9512                |
> > >
> > > Under reasonable binning values, the discretized outputs are still far from the one-sample-per-cell regime. Second, when binning becomes excessively fine, the occupancy does become much sparser, which is exactly why overly fine discretization is undesirable in practice.
> > >
> > > 2. Regarding label-only setting:
> > >
> > >
> > > We agree that a strict label-only output can indeed be represented as a one-hot vector. But it is much less informative than the prediction vector used in the current formulation.
> > >
> > > In the one-hot case, all samples predicted as the same class share the same T. This collapses the output representation from a rich confidence-bearing vector to only C possible states, removing ranking, margin, and uncertainty information. The issue is therefore not that label-only outputs cannot be represented, but that the resulting representation is far too coarse to preserve the main signal exploited by the current method.
> > >
> > >
> > > 3. Regarding $I(X;T)=H(T)$ in deterministic model
> > >
> > > This can simplify the practical computation, and we agree that the notation with this can be simplified to state more clearly. For the theorems and proofs, however, we do not think a rewrite is necessary, because they do not depend on this identity itself, but on the comparative behavior of the information-plane quantities between training and non-training data and the resulting statistical testing framework. So, in our view, this point mainly affects notation and implementation-level interpretation, rather than the substance of the theorems or proofs. We will revise the paper accordingly to make this distinction explicit.
> > >
> > > We hope these clarifications better address the reviewer’s concerns.

---

### Official Review · Reviewer_eLei · 2026-03-13

**Soundness:** 3
**Presentation:** 2
**Significance:** 2
**Originality:** 2
**Overall Recommendation:** 4
**Confidence:** 3

**Summary:**

This paper proposes DataGuard, a non-intrusive framework for auditing dataset misuse, where, given a black-box model, a dataset owner can determine whether their protected dataset was used to train the evaluated model. Unlike intrusive methods that require dataset modifications (e.g., watermarking), DataGuard leverages non-intrusive information forensics and a multivariate test to assess data usage within the model. It uses mutual information metrics based on feature information and label information to establish mutual information inequalities that distinguish training data from non-training data. These inequalities are then fed into the Hotelling $T^2$ test to obtain a $ p$-value. The authors provide extensive evaluations across architectures, partial usage scenarios, and defenses like DP-SGD.

**Compliance With Llm Reviewing Policy:**

Affirmed.

**Final Justification:**

The rebuttal substantially improves my confidence in the paper and weakens the main reasons for my initial weak rejection.

**Key Questions For Authors:**

-  Can you provide a direct finite-sample calibration study for the Hotelling $T^2$ test, beyond the asymptotic $χ_2^2$ argument?

- Can you better justify the scope of Theorem 3.3, or provide empirical evidence showing when its assumptions do and do not hold?

- How robust is the method to realistic mismatches between target and auxiliary datasets beyond class-proportion shift? If the authors can show robustness under broader and more realistic mismatches, e.g., covariate shift, label noise, curation differences, or temporal drift, that would strengthen the evaluation
- Based on B.4, how sensitive are the results to the mutual-information estimator with and its hyperparameters, including setting the number of intervals to 200?

**Limitations:**

the paper does discuss the main technical limitations.

**Strengths And Weaknesses:**

Strengths:
- The method avoids compromising model performance or introducing security risks associated with intrusive methods such as backdoors/watermarks.

- Their approach moves from heuristic signals (e.g., accuracy gaps) to a more principled analysis of training dynamics based on two interpretable information metrics with a multivariate test.

- The paper covers wide scenarios from positive and negative cases, stochastic training variation, partial exposure, mixed training, moderate distribution shift, DP-SGD, and SimCLR-based visual encoders.

- Authors show that the relative separation of information signals persists even when absolute values are suppressed by differential privacy.

Major Issues:

- A key limitation of the method is that it relies heavily on having auxiliary non-training datasets that are well-matched to the target domain. The test assumes that, under the null hypothesis, the target and auxiliary metric vectors come from the same distribution. While the paper does test a mild non-i.i.d. setting, I do not think this is enough. In realistic settings, auxiliary data may differ due to curation bias, time drift, label noise, or a lack of a larger reserve pool. Since the method is entirely based on comparing target and auxiliary datasets, even a moderate mismatch could be mistaken for evidence of training. I think this should be emphasized as a main limitation, and the paper would be stronger with broader stress tests covering more realistic forms of distribution mismatch. I appreciate that the paper does acknowledge the auxiliary-data issue: it notes the i.i.d. assumption and points readers to Section 5.1 for the non-i.i.d. case, and Appendix H explicitly evaluates non-i.i.d. auxiliary datasets by perturbing class proportions. However, the results also show that this mismatch increases false detections: FDR is 0 across all settings in Table 6 under the standard setup, but becomes nonzero in Table 7 under the non-i.i.d. setup, reaching as high as 10 for DeiT-TIN. This is important because it suggests that a distribution mismatch between target and auxiliary data can itself look like evidence of training.

- Theorem 3.3 proof is not fully convincing as the argument depends on a fairly idealized view of training dynamics, with strong assumptions that the model weights settle into a kind of stable distribution under the training objective, which may not match how modern neural networks are actually trained. Because of that, the result feels more like a useful intuition under specific conditions than a general theorem about training vs. non-training data. I would recommend softening the claim and framing it as a hypothesis supported by experiments. It would also help to add empirical checks showing whether the claimed behavior of I(X;T)still holds under different training setups (not only SGD) with Adam optimizer, data augmentation, momentum, learning-rate schedules, normalization, and early stopping, i.e., when compression is weak or early stopping is used.
	For the Hotelling’s test, it treats the target dataset as a single sample compared to a group of $k$ auxiliary datasets. Here, the "target" is only one point in the information plane, the covariance matrix used to calculate the $T^2$ statistic is derived entirely from the auxiliary data. If the auxiliary data is not perfectly i.i.d. or is too small (e.g., $k=100$), the resulting $p$-value can be numerically unstable.
	Table 1, Table 2, and Table 4 reported p-values are often either extremely saturated (<$10^{-308}$ ) or exactly 1.0. That is impressive, but it makes it difficult to judge whether the test is genuinely well calibrated or simply operating in a regime where the statistic is numerically extreme. Authors should report raw $T^2$ values, effect sizes, and confidence intervals over runs.
	The authors worked with a single sample at the dataset level, and the number of auxiliaries is not large. How accurate the claimed $χ_2^2$ approximation is in the actual regime used by the experiments?  Including QQ plots of $T^2$ against$ χ_2^2$, permutation tests, or bootstrap-based calibration


Minors:

- Several abbreviations are defined several times.
- Clarify exactly what Tis is in every experiment.
- Setting p=1when either inequality is violated is understandable as a conservative rule, but why is this the right correction, and whether other conservative choices were considered?

---

> ### Author Rebuttal · Authors · 2026-03-30
>
> We thank the reviewer for appreciating our principled analysis and broad evaluation. We have added new experiments per your comments. Additional figures are in https://www.dropbox.com/scl/fi/gv7urkwc64kphc01hyg7p/Additional-results.pdf?rlkey=7lh5z7wnt2y6egaftwls1zzx8&st=13g907pg&dl=0
>
> **Regarding Hotelling $T^2$:**
>
> We agree that the target metric is a single observation in the 2D plane, while the covariance is estimated from the auxiliary datasets, so finite-sample calibration is important. Thus, we added a finite-sample calibration study in the revision. Specifically, for different numbers of auxiliary datasets K, we empirically construct the null distribution of the $T^2$ statistic and compare it with the theoretical $\chi^2_2$ reference using QQ plots. The results in the supplementary link show reasonable agreement already for K=4-10, with the approximation becoming more stable as K grows.
>
> The extremely small p-values in positive cases arise because the target dataset is strongly separated from the auxiliary non-training baselines after training, leading to very large $T^2$ values. In contrast, in negative cases, both target and auxiliary datasets are non-training data, so the directional MI inequalities are often not consistently satisfied. As described in Section 3.4.2, whenever these inequalities are violated, we conservatively set p=1 to reduce false positives. Figure 3 illustrates this behavior. Besides, we view the raw Hotelling $T^2$ statistic itself as the most direct effect-size measure in this setting. Figures 2, 4, and 5 already report raw $T^2$ values for several important settings, and we will include more complete summaries and confidence intervals in the revised version.
>
>
> **Regarding Theorem 3.3:**
>
> We will reframe Theorem 3.3 as a theoretical intuition tied to the compression phase. Empirically, we further tested SGD with momentum and Adam on CIFAR-10 with ResNet-18 and VGG-16, and observed that the I(X;T) inequality already holds at epochs 20, 60, and 100 across four settings. For example, for ResNet18-Adam, the values are 9.04 vs. 9.25 at epoch 20, 5.90 vs. 6.80 at epoch 60, and 3.61 vs. 5.49 at epoch 100. This suggests that while absolute compression varies across optimizers and architectures, the relative MI difference still exists even under early stopping. Additional figures in the supplementary link further show this trend across epochs.
>
> **Regarding auxiliary datasets:**
>
> We agree that robustness to auxiliary mismatch is an important concern, so we added broader mismatch tests per your suggestions.
>
> For covariate shift, we constructed shifted auxiliary datasets by applying progressively stronger perturbations. The mild shift applies ColorJitter (brightness/contrast/saturation=0.1, hue =0.02); the strong setting applies heavier ColorJitter (brightness/contrast/saturation=0.3, hue =0.08), followed by GaussianBlur (kernel size  5, σ∈[0.5,1.5]) and RandomAffine (rotation up to $15^∘$). For label mismatch, we injected random label noise into the auxiliary datasets at rates of 10% and 20%. For reduced auxiliary pools, we removed 10% or 20% of auxiliary samples from the highest- and lowest-confidence tails according to the suspect model’s predictions, simulating realistic filtering effects during dataset curation.
>
> The CDR/FDR under 100 runs are shown below. Here C/F = CDR/FDR, LN = label noise, rm = sample removal
> Model| Mild C/F | Strong C/F | 10% LN C/F | 20% LN C/F | 10% rm C/F | 20% rm C/F
> -|-|-|-|-|-|-|
> VG-CF10|100/0| 99/5| 97/0 | 97/0|100/1| 98/0
> VG-CF100|100/0| 96/4| 97/1| 95/0|100/0| 97/4
> VG-TIN|99/2| 96/6| 97/2| 94/2| 95/2|92/5
> RN-CF10|100/0| 99/2| 99/0|100/1|100/0|99/2
> RN-CF100|100/0| 95/5| 98/2| 95/2|97/2| 97/2
> RN-TIN| 99/1|94/7| 95/4|92/0| 96/2| 94/5
> DeiT-CF10|100/0| 95/9|100/2|95/3|99/0|96/6
> DeiT-CF100|100/0| 95/8|97/5|96/4|98/0|95/5
> DeiT-TIN| 96/3| 92/11|95/2|92/6|92/3|92/8
>
> Overall, DataGuard remains reliable under mild mismatch, with high CDR and near-zero FDR. As mismatch becomes stronger, FDR increases and CDR decreases, especially in more challenging settings such as DeiT/Tiny-ImageNet. These results clarify that DataGuard is not intended to handle arbitrary auxiliary mismatch, but remains practically useful in its target regime, where reasonably representative non-training baselines can be constructed.
>
> **Regarding the MI estimator:**
>
> We agree that the MI estimator may be sensitive to its hyperparameters. To examine this sensitivity, we performed an ablation over the number of bins (50, 100, 200) and temperature scaling (0.5, 1, 2, 5). The auditing performance remains overall stable across these settings, suggesting that DataGuard is not tied to the specific choice of 200 bins. Due to the rebuttal word limit, the results are provided in the first table in our response for Reviewer 6bSw.
>
> We will add all these results to the final version. We hope these clarifications address the reviewer’s concern and support a more positive overall assessment.

---

> > ### Author Rebuttal · Reviewer_eLei · 2026-04-03
> >
> > I appreciate the detailed rebuttal and the additional experiments. The authors have meaningfully addressed several of my main concerns. My remaining reservation is that the final p-value procedure still appears to rely in part on the manual
> > $p=1$ safeguard. So the method is not yet a completely clean statistical test based on $𝑇^2$ alone.  The rebuttal substantially improves my confidence in the paper and weakens the main reasons for my initial weak rejection.

---

> > > ### Author Response · Authors · 2026-04-03
> > >
> > > We sincerely thank the reviewer for the careful reconsideration and for updating the score. We are very encouraged that the rebuttal and the additional experiments have meaningfully addressed your main concerns.
> > >
> > > Regarding the remaining reservation, we agree that the final p-value procedure is not a perfectly pure test based on the Hotelling $T^2$ statistic alone, because it includes the safeguard for logically contradictory cases. Our intention with this safeguard is not to artificially strengthen the result, but to make the procedure more conservative in practice. If the target dataset violates the expected information-plane direction relative to the auxiliary datasets, then it should not be treated as evidence of training usage, regardless of the quadratic statistic alone. This design is also consistent with our theoretical analysis, which gives us confidence that such a safeguard is appropriate for reducing false positives.
> > >
> > > More broadly, avoiding false positives is a central challenge in non-intrusive auditing, which is exactly why we approach the problem from a formal theoretical perspective and then add a conservative safeguard in the final decision rule. Empirically, the fact that the non-training cases consistently return $p=1.0$ is also consistent with the role of this safeguard and supports its practical reasonableness.
> > >
> > > We also agree that this distinction should be stated more clearly in the paper. In the final version, we will explicitly describe the safeguard as a conservative practical rule layered on top of the $T^2$-based test.
> > >
> > > We very much appreciate your positive reassessment and your constructive suggestion.

---

### Official Review · Reviewer_MB8L · 2026-03-13

**Soundness:** 3
**Presentation:** 3
**Significance:** 2
**Originality:** 2
**Overall Recommendation:** 4
**Confidence:** 3

**Summary:**

This manuscript outlines a significant problem: checking whether a dataset was used to train a model without modifying the data itself. Overall, the manuscript assesses a central concept by proposing DataGuard, a non-intrusive auditing method that compares model behavior on a target dataset and auxiliary datasets using mutual information and statistical testing. The experiments show that the method can detect dataset usage in controlled settings and performs well across several image models.

**Compliance With Llm Reviewing Policy:**

Affirmed.

**Key Questions For Authors:**

no questions

**Limitations:**

yes

**Strengths And Weaknesses:**

Strengths:
This paper studies an important and practical problem in dataset auditing, presenting a method that does not require modifying the original dataset.
The paper includes both theoretical analysis and extensive experiments and the results are consistent across several models and datasets.

Weaknes:
A limitation is that the method depends on how representative the auxiliary datasets are, since weak or mismatched auxiliary data may affect the reliability of the auditing results.

---

> ### Author Rebuttal · Authors · 2026-03-30
>
> We thank the reviewer for the positive assessment and for recognizing the practical importance of non-intrusive dataset auditing, as well as the consistency of our results across models and datasets.
>
> **Regarding the concern about auxiliary datasets:**
>
> We agree that the quality of auxiliary datasets is an important scope condition of DataGuard. To better assess this issue, we added more comprehensive auxiliary-mismatch tests covering covariate shift, label noise, and reduced auxiliary pools.
>
> For covariate shift, we constructed shifted auxiliary datasets by applying progressively stronger perturbations to the auxiliary baselines. Specifically, the mild shift applies ColorJitter with brightness, contrast, and saturation set to 0.1 and hue set to 0.02. The strong shift applies ColorJitter with brightness, contrast, and saturation set to 0.3 and hue set to 0.08, followed by GaussianBlur with kernel size 5 and σ∈[0.5,1.5], and RandomAffine with rotation up to 15∘.
>
> For label mismatch, we injected random label noise (i.e., wrong label) into the auxiliary datasets at rates of 10% and 20%. This simulates imperfect annotation quality.
>
> For reduced auxiliary pools, we constructed auxiliary datasets by removing 10% or 20% of the most- and least-confident samples predicted by the suspect model, simulating realistic differences caused by filtering and quality control.
>
> The CDR/FDR results over 100 runs are shown below. Here, C/F = CDR/FDR, LN = label noise, and rm = sample removal.
> Model| Mild C/F | Strong C/F | 10% LN C/F | 20% LN C/F | 10% rm C/F | 20% rm C/F
> -|-|-|-|-|-|-|
> VG-CF10|100/0| 99/5| 97/0 | 97/0|100/1| 98/0
> VG-CF100|100/0| 96/4| 97/1| 95/0|100/0| 97/4
> VG-TIN|99/2| 96/6| 97/2| 94/2| 95/2|92/5
> RN-CF10|100/0| 99/2| 99/0|100/1|100/0|99/2
> RN-CF100|100/0| 95/5| 98/2| 95/2|97/2| 97/2
> RN-TIN| 99/1|94/7| 95/4|92/0| 96/2| 94/5
> DeiT-CF10|100/0| 95/9|100/2|95/3|99/0|96/6
> DeiT-CF100|100/0| 95/8|97/5|96/4|98/0|95/5
> DeiT-TIN| 96/3| 92/11|95/2|92/6|92/3|92/8
>
> Overall, DataGuard remains reliable under mild mismatch, with consistently high CDR and near-zero FDR. As the mismatch becomes stronger, FDR increases and CDR decreases, especially in more challenging settings such as DeiT/Tiny-ImageNet. These results clarify that DataGuard is not intended to be robust to arbitrary auxiliary mismatch, but remains practically useful in the regime it targets: when reasonably representative non-training baselines can be constructed.
>
> We will add these results and revise the paper to present auxiliary representativeness more explicitly as a scope condition of the framework, while also highlighting practical scenarios where such baselines naturally exist. More broadly, we view this work as a meaningful first step toward principled non-intrusive dataset auditing, and we hope it motivates future work on auxiliary-free or synthetic-baseline settings.
>
> **Regarding the broader contribution:**
>
> More importantly, we view the theoretical component not just as an additional analysis, but as a shift in how dataset auditing is formulated: away from heuristic attack-based verification and toward a principled, statistically grounded auditing framework. We believe this shift is especially meaningful for a serious problem like dataset auditing, where the outcome may carry compliance or legal implications.
>
> We hope these clarifications help address the reviewer’s concern.

---

### Decision · Program_Chairs · 2026-04-30

**Decision:**

Accept (regular)

**Comment:**

While the reviewers agree that the paper proposes an important non-intrusive framework (called DataGuard) for auditing dataset misuse with good theoretical analyses and experiments, there are also concerns on the reliance on representative auxiliary datasets, theoretical guarantees, other technical details, experiments, and presentation.

During the rebuttal, the authors addressed most of the concerns sufficiently. Reviewer GyDQ did have a remaining concern about using a high \epsilon parameter for a DP-SGD experiment, but the authors responded that DataGuard retrains useful auditing signals under moderate noisy training with new experiments. In the end, the reviewer did not provide a Final Justification, so the AC assumes the reviewer does not disagree with the response.

Given that there are three Weak Accepts and one Weak Reject that was given because of the high \epsilon concern above, the recommendation is to accept the paper.